# Partial Information Decomposition via Normalizing Flows in Latent Gaussian Distributions

**Wenyuan Zhao**[1]    **Adithya Balachandran**[2]    **Chao Tian**[1]    **Paul Pu Liang**[2]

[1]Texas A&M University
[2]Massachusetts Institute of Technology
[1]{wyzhao, chao.tian}@tamu.edu, [2]{adithyab, ppliang}@mit.edu

## Abstract

The study of multimodality has garnered significant interest in fields where the analysis of interactions among multiple information sources can enhance predictive modeling, data fusion, and interpretability. Partial information decomposition (PID) has emerged as a useful information-theoretic framework to quantify the degree to which individual modalities independently, redundantly, or synergistically convey information about a target variable. However, existing PID methods depend on optimizing over a joint distribution constrained by estimated pairwise probability distributions, which are costly and inaccurate for continuous and high-dimensional modalities. Our first key insight is that the problem can be solved efficiently when the pairwise distributions are multivariate Gaussians, and we refer to this problem as Gaussian PID (GPID). We propose a new gradient-based algorithm that substantially improves the computational efficiency of GPID based on an alternative formulation of the underlying optimization problem. To generalize the applicability to non-Gaussian data, we learn information-preserving encoders to transform random variables of arbitrary input distributions into pairwise Gaussian random variables. Along the way, we resolved an open problem regarding the optimality of joint Gaussian solutions for GPID. Empirical validation in diverse synthetic examples demonstrates that our proposed method provides more accurate and efficient PID estimates than existing baselines. We further evaluate a series of large-scale multimodal benchmarks to show its utility in real-world applications of quantifying PID in multimodal datasets and selecting high-performing models.

## 1 Introduction

Multimodal machine learning is a fast-growing subarea of artificial intelligence research that aims to develop systems capable of integrating and fusing many heterogeneous modalities [2, 31]. In addition to much empirical progress, there has been a recent drive toward building theoretical foundations to understand when information in individual modalities is important and how this information becomes contextualized in the presence of other modalities [32]. This aspect of quantifying interactions provides valuable insights into the significance of different modalities, the necessary amount of data required in each modality, and the methods most suitable for fusing multimodal representations [15, 18, 26, 27, 35, 61]. Partial Information Decomposition (PID), an advanced framework rooted in information theory, has been used as a formal framework to analyze how information is distributed among multiple data sources for a target task variable [5, 17, 55].

One fundamental challenge of PID is its high computational complexity, especially when the size and dimensionality of the datasets are large [43]. Accurate estimation of information-theoretic measures, such as mutual information (MI), from empirical data is non-trivial and even prohibitive, particularly for high-dimensional or continuous distributions [38]. Estimating PID also presents a significant

39th Conference on Neural Information Processing Systems (NeurIPS 2025).

challenge, as analytic approximations of these quantities can only be obtained when the features are pointwise discrete or low-dimensional. However, for large-scale datasets, the number of optimization variables can be exponential in the number of neurons [53]. The computational burden requires significant approximations or simplifications based on sampling that may compromise the accuracy of PID estimation [12]. Therefore, it is necessary to develop a PID estimator that applies to continuous and high-dimensional multimodal data.

In this paper, we identify the first key insight that PID for Gaussian distributions, known as Gaussian PID (GPID), is scalable for continuous modalities. We first develop a new algorithm for GPID, which is exact and efficient in multivariate Gaussian distributions with high dimensionality. The second insight is to learn feature encoders to transform arbitrary input distributions into a latent Gaussian space, without violating the information interactions between the original modalities. This transformation can be done by normalizing flows [39], whose invertible bijections preserve information while bringing the joint distributions closer to Gaussians [7]. Subsequently, the PID estimate can be dramatically simplified in latent Gaussian distributions, which enjoy a closed-form analysis of differential entropy and MI. We summarize the following contributions in this paper:

1. We propose a new gradient-based algorithm, called Thin-PID, which significantly enhances the computational efficiency of PID estimates in latent Gaussian distributions.
2. We develop a new framework, called Flow-PID, to generalize Gaussian PID algorithms to arbitrary input modalities using flow-based information-preserving encoders.
3. We demonstrate the improved accuracy and efficiency of our proposed Thin-PID and Flow-PID on diverse synthetic datasets with ground truth information labels. Further evaluation on a series of large-scale multimodal benchmarks shows its utility in real-world applications.

Finally, we release the data and code for Thin-PID and Flow-PID to encourage further studies of multimodal information and modeling at `https://github.com/warrenzha/flow-pid`.

## 2  Background and related work

Let $X_1, X_2, Y$ be three random vectors of dimension $d_{X_1}, d_{X_2}, d_Y$ in their respective alphabets $\mathcal{X}_1$, $\mathcal{X}_2$, and $\mathcal{Y}$. Denote $\Delta$ as the set of joint distributions in $(\mathcal{X}_1, \mathcal{X}_2, \mathcal{Y})$. In multimodal learning, we are concerned with inferring on the class label or regression value $Y$, whether the modalities $X_1$ and $X_2$ can provide useful information together, beyond their unique information, and how much common information for $Y$ we can ascribe to both $X_1$ and $X_2$.

### 2.1  Partial information decomposition

PID decomposes the total information $I_p(X_1, X_2; Y)$, where $I_p$ denotes the *Shannon mutual information* [9] in joint distribution $p$, between the target $Y$ and two basic features $(X_1, X_2)$ into four non-negative parts [55]:

$$I_p(X_1, X_2; Y) = U_1(Y; X_1 \backslash X_2) + U_2(Y; X_2 \backslash X_1) + R(Y; X_1, X_2) + S(Y; X_1, X_2), \quad (1)$$

where the four terms on the right-hand side respectively represent the information regarding $Y$ that is uniquely in $X_1$, uniquely in $X_2$, redundantly in either $X_1$ or $X_2$, synergistically in both $X_1$ and $X_2$. It is also often required that the decomposition satisfy the conditions on individual information: $I_p(Y; X_1) = R(Y; X_1, X_2) + U_1(Y; X_1 \backslash X_2)$, $I_p(Y; X_2) = R(Y; X_1, X_2) + U_2(Y; X_2 \backslash X_1)$.

The decomposition is not unique, since we only have three linear equations to specify four variables. We adopt the definition of PID introduced by Bertschinger et al. [5], which appears to align effectively with multimodal learning applications and has been shown to facilitate model selection [27].

**Definition 2.1** (PID). The redundant, unique, and synergistic information are given by

$$R = \max_{q \in \Delta_p} I_q(X_1; X_2; Y), \quad (2)$$

$$U_1 = \min_{q \in \Delta_p} I_q(X_1; Y | X_2), \quad U_2 = \min_{q \in \Delta_p} I_q(X_2; Y | X_1), \quad (3)$$

$$S = I_p(X_1, X_2; Y) - \min_{q \in \Delta_p} I_q(X_1, X_2; Y), \quad (4)$$

where $\Delta_p := \{q \in \Delta : q(x_i, y) = p(x_i, y), \forall y \in \mathcal{Y}, x_i \in \mathcal{X}_i, i \in [2]\}$, and $I_q$ is the mutual information (MI) over the joint distribution $q(x_1, x_2, y)$. Note that $\Delta_p$ only preserves the marginals $p(x_1, y)$ and $p(x_2, y)$, but not necessarily the joint distribution $p(x_1, x_2, y)$.

## 2.2 Normalizing flows

Estimating MI and differential entropy in PID can be very challenging for high-dimensional data, unless in multivariate Gaussian distributions. Normalizing flows are a class of machine learning models that are used to transform a simple, tractable probability distribution (e.g. Gaussian distributions) to a more complex distribution, while preserving exact likelihood computation and invertibility [39]:

$$x = f(z) := f_k \circ f_{k-1} \circ \cdots \circ f_1(z), \tag{5}$$

where $f(\cdot)$ is *invertible* and *differentiable*, $z = f^{-1}(x)$ and $p_X(x) = p_Z(z) \left| \det \frac{\partial z}{\partial x} \right|^{-1}$.

Unlike generative modeling, we adopt the inverse process of normalizing flows in PID to transform complex input modalities in such a way that MI can be preserved and computed using latent representations $I_p(f_X(X); f_Y(Y)) = I_p(X; Y)$. Subsequently, information interactions are computed efficiently in latent Gaussian distributions. This motivates us to design a novel framework for PID estimators by training normalizing flows that preserve the total information $I_p(X_1, X_2; Y)$ and transform input modalities into latent spaces that are well approximated by multivariate Gaussian representations.

## 2.3 Related work

**PID estimation**: There have been some recent efforts on PID estimators. Liang et al. [27] introduced the CVX estimator, which formulates the PID definition as a convex optimization problem that can be solved if $X_1$ and $X_2$ are discrete and small. Liang et al. [27] also developed the BATCH algorithm, which approximates PID for large datasets by parameterizing the joint distribution $q$ with neural networks and normalization to satisfy marginal constraints. Therefore, several of these methods are limited in scope and do not scale well to continuous, high-dimensional data. Venkatesh et al. [52] introduced $\sim_G$-PID, which restricts $q(x_1, x_2, y) \in \Delta_p$ to joint Gaussian distributions, where PID values are easier to estimate given the analytical entropy and MI. However, the optimality of joint Gaussian distributions remains an open question, which implies that the joint Gaussian solution may not be true in general and only provides an optimizing bound for GPID estimation.

**Information theory estimation**: Neural estimators of information-theoretic quantities have become essential tools in machine learning due to their ability to scale to high-dimensional, continuous data. Mutual Information Neural Estimator (MINE) leverages the Donsker–Varadhan representation of the KL divergence to construct a variational lower bound on MI [4]. Alternative estimators based on $f$-divergences, including NWJ [36] and InfoNCE [37], provide tighter bounds in specific cases. Entropy estimation has also been addressed via neural score matching and normalizing flows [44]. More recent works include [7, 16, 22]. These methods optimize variational bounds via gradient descent, which is parameterized by neural networks [40]. Such approaches cannot be directly adapted for PID estimation due to the additional optimization problem in Equations (2) to (4).

**Information-theoretic multimodal learning**: Estimating multimodal information has been a critical step toward developing better benchmarks and algorithms for multimodal learning. Several benchmarks are categorized by the types of information that modalities contribute, which subsequently inspire research into new multimodal fusion methods [30]. Deeper studies of multimodal information have also inspired new ways to guide the collection of pre-training data [6] and new multimodal pre-training objectives [34]. Multimodal contrastive learning is a popular approach in which representations of the same concept expressed in different modalities are matched together (i.e., positive pairs) and those of different concepts are far apart (i.e., negative pairs) [13, 25, 41]. It can be shown that contrastive learning provably captures redundant information across the two views [47, 48], and recent work has proposed extensions to capture unique and synergistic information [11, 28].

## 3 A new Gaussian PID theory and algorithm

In this section, we present a new PID estimator for Gaussian distributions and establish the notation set we use in the rest of the paper. The first contribution of this paper is to show that the optimal solution of the restricted PID optimization is jointly Gaussian. Secondly, we propose a new algorithm for GPID, which significantly enhances computational efficiency for high-dimensional features.

**Definition 3.1** (GPID). Let $\Delta^{\mathrm{G}}$ be the set of joint distributions, where $p(x_1, y)$ and $p(x_2, y)$ are pairwise Gaussian. The synergistic information $S$ about $Y$ in $X_1$ and $X_2$ is given by

$$S = I_p(X_1, X_2; Y) - \min_{q \in \Delta_p^{\mathrm{G}}} I_q(X_1, X_2; Y), \tag{6}$$

where $\Delta_p^{\mathrm{G}} := \{q \in \Delta^{\mathrm{G}} : q(x_i, y) = p(x_i, y), \forall y \in \mathcal{Y}, x_i \in \mathcal{X}_i, i \in [2]\}$. Subsequently, redundant and unique information can be computed by $R = \max_{q \in \Delta_p^{\mathrm{G}}} I_q(X_1; X_2; Y)$ and $U_1 = \min_{q \in \Delta_p^{\mathrm{G}}} I_q(X_1; Y | X_2)$, $U_2 = \min_{q \in \Delta_p^{\mathrm{G}}} I_q(X_2; Y | X_1)$ respectively.

GPID is exactly the PID problem when the pairwise marginals are known to be Gaussians. We adopt the following notation in the subsequent discussion. Suppose $[X^\top, Y^\top]^\top$ are jointly Gaussian vectors. Without loss of generality, we assume that they have zero mean and use the following notation to denote the covariance matrices. More detailed derivation can be found in Appendix A.

- $\Sigma_{XY}$ denotes the $(d_X + d_Y) \times (d_X + d_Y)$ auto-covariance matrix of the vector $[X^\top, Y^\top]^\top$.
- $\Sigma_{XY}^{\mathrm{off}}$ denotes the $d_X \times d_Y$ cross-covariance matrix (off-diagonal of $\Sigma_{XY}$) between $X$ and $Y$.

### 3.1 Optimality of joint Gaussian solution

In [52], instead of solving GPID, the authors directly restricted the set $\Delta_p^{\mathrm{G}}$ in GPID by placing the additional constraint that $q(x_1, x_2, y)$ is Gaussian, and then optimized within this restricted set, without showing that doing so does not cause any loss of optimality. In other words, $\sim_G$-PID may only provide a lower bound of $S$ for GPID in general.

> *Question: Is joint Gaussian solution $q(x_1, x_2, y)$ indeed optimal for GPID?*
> *Answer: Yes. It is optimal as long as $p(x_1, y)$ and $p(x_2, y)$ are Gaussians.*

**Optimality:** We next show that there always exists a jointly Gaussian optimizer $q(x_1, x_2, y)$ for GPID in Definition 3.1. Recall that we are interested in the following optimization problem:

$$\min_{q \in \Delta_p^{\mathrm{G}}} I_q(X_1, X_2; Y) = \min_{q \in \Delta_p^{\mathrm{G}}} \{h_q(Y) - h_q(Y | X_1, X_2)\}, \tag{7}$$

where $h_q(\cdot)$ denotes the differential entropy of the distribution $q$. Since the marginal $q(y)$ is preserved to be the same as in both $p(x_1, y)$ and $p(x_2, y)$, the problem is clearly equivalent to:

$$\max_{q \in \Delta_p^{\mathrm{G}}} h_q(Y | X_1, X_2). \tag{8}$$

The key to establishing the optimality of Gaussian solutions is the inequality given in Lemma 3.2, whose proof is given in Appendix A.1. It now only remains to argue that the upper bound on the right-hand side of (9) can be achieved with a jointly Gaussian $\hat{q} \in \Delta_p^{\mathrm{G}}$ for any $q \in \Delta_p^{\mathrm{G}}$. This is obvious since $\hat{q}$ has the same first and second moments as $q(x_1, x_2, y)$, and we have $q \in \Delta_p^{\mathrm{G}}$. Therefore, it is without loss of optimality to restrict the set $\Delta_p^{\mathrm{G}}$ to only joint Gaussians.

**Lemma 3.2.** *For any $q(x_1, x_2, y)$ with finite first and second moments, we have*

$$h_q(Y | X_1, X_2) \le h_{\hat{q}}(Y | X_1, X_2), \tag{9}$$

*where $\hat{q}(x_1, x_2, y)$ is a jointly Gaussian distribution with the same first and second moments as $q(x_1, x_2, y)$.*

### 3.2 Thin-PID: a new algorithm for GPID

We are now in a position to discuss the problem of how to compute GPID. Recall that Venkatesh and Schamberg [53] used a two-user Gaussian *broadcast channel* to interpret GPID, where $Y$ is the transmitter input variable, and $X_1, X_2$ are the channel outputs at the two individual receivers:

$$X_1 = H_1 Y + n_1 \text{ and } X_2 = H_2 Y + n_2. \tag{10}$$

Without loss of generality, we assume that $n_1$ and $n_2$ are independent additive white noise, i.e., $\Sigma_{n_1} = \Sigma_{X_1|Y} = I_{d_{X_1}}$ and $\Sigma_{n_2} = \Sigma_{X_2|Y} = I_{d_{X_2}}$. Otherwise, we can perform receiver-side linear transformations through eigenvalue decomposition. For the same reason, we can assume

Table 1: Complexity analysis of different GPID algorithms. The complexities of ED and SVD are cubic in the values shown in the table. Thin-PID achieves better complexity on any scale of computation.

|  | ED | SVD | Lin-Eqn Solve | Mul |
|---|---|---|---|---|
| Thin-PID | – | $\min(d_{X_1}, d_{X_2})$ | $2 * \min(d_{X_1}, d_{X_2})$ | $4 * \min(d_{X_1}, d_{X_2})$ |
| Tilde-PID | $d_{X_1} + d_{X_2}$ | $\max(d_{X_1}, d_{X_2})$ | $2 * (d_{X_1} + d_{X_2})$ | $8 * (d_{X_1} + d_{X_2})$ |

$Y \sim \mathcal{N}(0, \Sigma_Y)$ has zero-mean. Although the constraints on $p(x_1, y)$ and $p(x_2, y)$ can be specified by $\Sigma_{X_1 Y}$ and $\Sigma_{X_1 Y}$ (or equivalent $\Sigma_{n_1}$ and $\Sigma_{n_2}$)[1], the joint distribution of $n_1$ and $n_2$ for $q(X_1, X_2, Y)$ remains to be optimized. However, they can be assumed to be jointly Gaussian with covariance $\Sigma_{n_1 n_2}$ as shown in Section 3.1.

Using the interpretation of the Gaussian broadcast channel, synergistic information $S$ for GPID can be recast as the cooperative gain with the input signal $Y$ [46]. As a result, $\min_{q \in \Delta_p^G} I_q(X_1, X_2; Y)$ in $S$ can be determined by optimizing the worst possible correlation between $n_1$ and $n_2$ in the least favorable noise problem [58].

**Theorem 3.3** (Thin-PID). *The optimization problem of synergistic information $S$ in Definition 3.1 can be recast as minimizing the following objective function:*

$$\text{minimize: } \mathcal{L}\left(\Sigma_{n_1 n_2}^{off}\right) = \log \frac{\left| H \Sigma_Y H^\top + \Sigma_{n_1 n_2} \right|}{\left| \Sigma_{n_1 n_2} \right|}, \tag{11}$$

$$\text{subject to: } \Sigma_{n_1} = \Sigma_{n_2} = I, \ \Sigma_{n_1 n_2} \succeq 0,$$

*where $H := [H_1^\top, H_2^\top]^\top$ is the concatenation of two channel matrices, and $\Sigma_{n_1 n_2}$ is the covariance of two joint noise vectors $n_1$ and $n_2$ with cross-covariance $\Sigma_{n_1 n_2}^{off}$.*

In this objective, $H$ and $\Sigma_Y$ are constants that can be estimated directly from the marginals and will not affect the optimization. Similarly, $\Sigma_{n_1}$ and $\Sigma_{n_2}$, which are *diagonal* block matrices of sizes $d_{X_1} \times d_{X_1}$ and $d_{X_2} \times d_{X_2}$, can be whitened as identity matrices before optimizing GPID. Therefore, the only variable to be optimized is $\Sigma_{n_1 n_2}^{off}$, which is the *off-diagonal* block matrix of $\Sigma_{n_1 n_2}$.

**Proposition 3.4** (Projected gradient descent for Thin-PID). *The **gradient** of the unconstrained objective $\mathcal{L}(\Sigma_{n_1 n_2}^{off})$ in Equation (11) with respect to $\Sigma_{n_1 n_2}^{off}$ is given by*

$$\nabla \mathcal{L}\left(\Sigma_{n_1 n_2}^{off}\right) = -G_{1,1}^{-1} G_{1,2} B^{-1} + \Sigma_{n_1 n_2}^{off}\left(I - \Sigma_{n_1 n_2}^{off \top} \Sigma_{n_1 n_2}^{off}\right)^{-1}, \tag{12}$$

*where $B = G_{2,2} - G_{1,2}^\top G_{1,1}^{-1} G_{1,2}$, and we define the block matrix as*

$$G := \begin{bmatrix} G_{1,1} & G_{1,2} \\ G_{1,2}^\top & G_{2,2} \end{bmatrix} = H \Sigma_Y H^\top + \Sigma_{n_1 n_2}, \ G_{1,2} \in \mathbb{R}^{d_{X_1} \times d_{X_2}}. \tag{13}$$

*The **gradient descent** updates on $\Sigma_{n_1 n_2}^{off}$: $\Sigma_{n_1 n_2}^{off} \leftarrow RProp(\Sigma_{n_1 n_2}^{off})$ [42].*

*The **projection operator** onto the constraint set can be obtained by taking the singular value decomposition (SVD) on $\Sigma_{n_1 n_2}^{off}$:*

$$SVD\left(\Sigma_{n_1 n_2}^{off}\right) = U \Lambda V^\top, \tag{14}$$

$$Proj(\Sigma_{n_1 n_2}^{off}) \leftarrow U \bar{\Lambda} V^\top, \tag{15}$$

*where $\Lambda := diag(\lambda_i)$, $\bar{\lambda}_i := \min\left(\max\left(0, \lambda_i\right), 1\right)$ and $\bar{\Lambda} := diag(\bar{\lambda}_i)$.*

**Complexity analysis:** In **Thin-PID**, the computational bottleneck comes from determining SVD and inverting matrices. SVD on $\Sigma_{n_1 n_2}^{off}$ requires $O\left(\min(d_{X_1}, d_{X_2})^3\right)$ complexity. For the inverse matrices, note that $G_{1,1}^{-1}$ is constant and only needs to be computed once. The other inverse can be computed by solving linear equations with complexity $O(d_{X_2}^3)$. Without loss of generality, we

---

[1]The joint covariance $\Sigma_{X_i Y}$ can be fully specified by $\Sigma_{X_i|Y} = \Sigma_{n_i}$ since $\Sigma_Y$ is constant which can be directly computed from target $Y$.

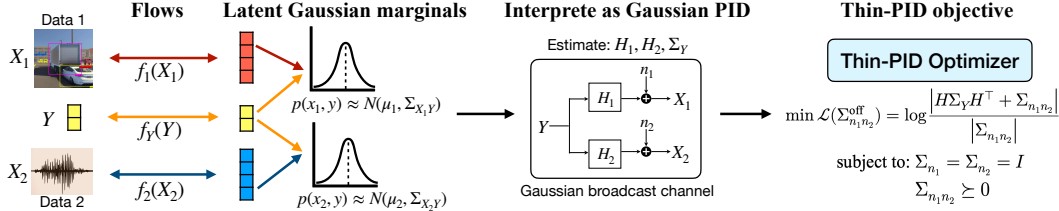

Figure 1: Flow-PID learns latent Gaussian encoders, parameterized by the Cartesian flow $f_1 \times f_2 \times f_Y$, to transform input modalities $(X_1, X_2, Y)$ into Gaussian marginal distributions. Then, PID values can be computed efficiently via Thin-PID under the equivalent interpretation of GPID.

assume $d_{X_1} \geq d_{X_2}$ since we can always exchange input modalities. In contrast, the state-of-the-art **Tilde-PID**[2] proposed by Venkatesh et al. [52] requires the eigenvalue decomposition (ED) on $\Sigma_{n_1 n_2}$ with the dominant complexity $O\left((d_{X_1} + d_{X_2})^3\right)$. As shown in Table 1, Thin-PID achieves significant improvement in computational efficiency, especially when the feature dimensions are high and $d_{X_1} \gg d_{X_2}$. More detailed complexity analysis is shown in Appendix A.4.

## 4 Learning a latent Gaussian encoder via normalizing flows

When the marginal distributions $p(x_1, y)$ and $p(x_2, y)$ are not Gaussian, computing PID from data samples requires estimating the joint distribution in some manner. We propose a novel approach where we learn a feature encoder transforming $(X_1, X_2, Y)$ into a latent space, such that they are well-approximated by Gaussian marginal distributions, then utilizing Thin-PID to perform the computation in an efficient way when modalities are continuous and high-dimensional.

Let $\hat{X}_1 = f_1(X_1)$, $\hat{X}_2 = f_2(X_2)$, $\hat{Y} = f_Y(Y)$ be three transformations that are defined by three neural networks, respectively. Given the dataset $\mathcal{D} = \{(x_1^{(j)}, x_2^{(j)}, y^{(j)}), j = 1, 2, \ldots, N\}$, ideally the transformations should satisfy the conditions: 1) The transformations are invertible such that the MI does not change; 2) The marginal distributions $p(\hat{x}_1, \hat{y})$ and $p(\hat{x}_2, \hat{y})$ are well-approximated by Gaussian distributions. Our goal is to learn a Cartesian product of normalizing flows $f_1 \times f_2 \times f_Y$ which will preserve the total mutual information as $I_{\hat{p}}(f_1(X_1), f_2(X_2); f_Y(Y)) = I_p(X_1, X_2; Y)$ according to Theorem 4.1. We refer to this method as Flow-PID, illustrated in Figure 1.

**Theorem 4.1** (Invariance of total MI under bijective mappings). *Let $X_1, X_2, Y$ be absolutely continuous vectors, $f_1 : \mathcal{X}_1 \to \mathbb{R}^{d_{X_1}}$, $f_2 : \mathcal{X}_2 \to \mathbb{R}^{d_{X_2}}$ and $f_Y : \mathcal{Y} \to \mathbb{R}^{d_Y}$ be bijective piecewise smooth mappings with tractable Jacobians. Then $I_{\hat{p}}(f_1(X_1), f_2(X_2); f_Y(Y)) = I_p(X_1, X_2; Y)$.*

**Corollary 4.2.** *Under the same conditions in Theorem 4.1, the PID of $(f_1(X_1), f_2(X_2), f_Y(Y))$ is the same as the PID of $(X_1, X_2, Y)$.*

**Corollary 4.3.** *Let $(X, Y)$ be absolutely continuous with PDF $p(x, y)$. Let $q(x, y)$ be a PDF defined in the same space as $p(x, y)$. Then $|I_p(X; Y) - I_q(X; Y)| \leq KL(p(x, y) \| q(x, y))$.*

### 4.1 Gaussian marginal loss

Theorem 4.1 and Corollary 4.2 imply that PID can be solved in a latent space through invertible transformations that preserve the total information. However, it could restrict the possible distributions to an unknown family. Instead, we approximate the PDF in latent space via variational Gaussian marginals $q(x_1, y)$ and $q(x_1, y)$ with tractable pointwise MI, and train $q$ and $f_1 \times f_2 \times f_Y$ to minimize the discrepancy between the real and the approximated total MI.

To regularize the Cartesian flows with Gaussian marginal distributions, we simultaneously minimize $\mathrm{KL}(p(\hat{x}_1, \hat{y}) \| \mathcal{N}(\mu_1, \Sigma_{X_1 Y}))$ and $\mathrm{KL}(p(\hat{x}_2, \hat{y}) \| \mathcal{N}(\mu_2, \Sigma_{X_2 Y}))$, where $\Sigma_{X_1 Y}$ and $\Sigma_{X_2 Y}$ are the covariance of the variational Gaussian marginals, and $\mathrm{KL}(\cdot \| \cdot)$ denotes the KL divergence. Note that maximizing the likelihood of $f_1(X_1) \times f_Y(Y)$ and $f_2(X_2) \times f_Y(Y)$ also minimizes $\mathrm{KL}\left(p_{X_1 Y} \circ (f_1^{-1} \times g^{-1}) \| \mathcal{N}(\mu_1, \Sigma_{X_1 Y})\right)$ and $\mathrm{KL}\left(p_{X_2 Y} \circ (f_2^{-1} \times g^{-1}) \| \mathcal{N}(\mu_2, \Sigma_{X_2 Y})\right)$.

---

[2]We refer to $\sim_G$-PID as Tilde-PID in the sequel.

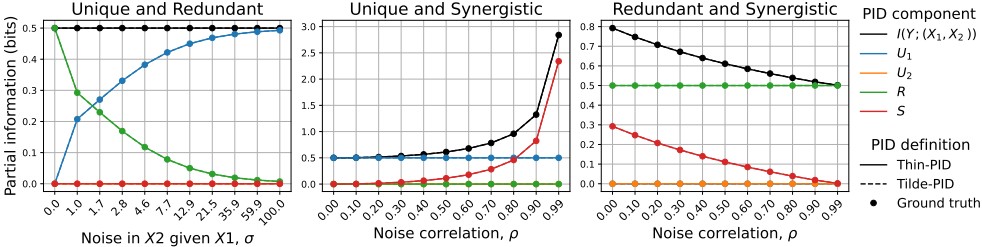

Figure 2: PID values for 1D Gaussian example with different types of interactions. Thin-PID and Tilde-PID agree exactly with the ground truth.

Therefore, the Gaussian marginal regularizer is equivalent to maximize the log-likelihood of $\{(x_i^{(j)}, y^{(j)})\}_{j=1}^N$ such that $x_i^{(j)} = f_i^{-1}(\hat{x}_i^{(j)})$, $y^{(j)} = g^{-1}(\hat{y}^{(j)})$, where $(\hat{x}_i^{(j)}, \hat{y}^{(j)})$ are sampled from variational Gaussian distribution $\mathcal{N}(\mu_i, \Sigma_{X_i Y})$.

**Proposition 4.4** (Gaussian marginal loss for Flow-PID). *Given data samples $\{(x_1^{(j)}, x_2^{(j)}, y^{(j)})\}$, the Gaussian marginal objective for $p(\hat{x}_i, \hat{y})$ is given by*

$$\mathcal{L}_{\mathcal{N}}(x_i^{(j)}, y^{(j)}) = \mathcal{L}_{\mathcal{N}(\mu_i, \Sigma_{X_i Y})}(\hat{x}_i^{(j)}, \hat{y}^{(j)}) + \log\left|\det\frac{\partial f_1(x_i)}{\partial x_i}\right| + \log\left|\det\frac{\partial f_Y(y)}{\partial y}\right|, \qquad (16)$$

*where $\mu_i$ and $\Sigma_{X_i Y}$ are determined by the Gaussian maximum likelihood. The objective function of the latent Gaussian encoder is*

$$\mathcal{L}_{flow}(\{X_1, X_2, Y\}) = \mathcal{L}_{\mathcal{N}}(\{(X_1, Y)\}) + \mathcal{L}_{\mathcal{N}}(\{(X_2, Y)\}). \qquad (17)$$

## 5 Synthetic PID validation

In this section, we validate our **Thin-PID** and **Flow-PID** on synthetic Gaussian and non-Gaussian examples with known ground truth, compared with **Tilde-PID** designed for GPID [52], **CVX**, and **BATCH** designed for features with discrete support [27]. Details of experimental settings are provided in Appendix C.

### 5.1 Validating Thin-PID on canonical Gaussian examples

**1D broadcast channel**: We first validate the accuracy of Thin-PID on canonical Gaussian examples with $d_{X_1} = d_{X_2} = d_Y = 1$. Let $Y \sim \mathcal{N}(0, 1)$. We design 3 cases with: 1) *unique* and *redundant*; 2) *unique* and *synergistic*; 3) *redundant* and *synergistic* information. Detailed settings can be found in Appendix C.1. The ground truth can be solved exactly by MMI-PID [3] when $p(x_1, x_2, y)$ is 1D Gaussian. The degrees of interactions estimated by Thin-PID and Tilde-PID are shown in Figure 2. We observe that Thin-PID exactly recovers the ground truth in all three canonical Gaussian cases, which corroborates the correctness of our proposed Thin-PID algorithm for GPID.

**GPID examples at higher dimensionality**: (i) **Cooperative Gain**: Let $X_{1,1} = \alpha Y_1 + n_{1,1}$, $X_{2,1} = Y_1 + n_{2,1}$, $X_{1,2} = Y_2 + n_{1,2}$, $X_{2,2} = 3Y_2 + n_{2,2}$, where $Y_1, Y_2, n_{1,i}, n_{2,i} \sim$ i.i.d. $\mathcal{N}(0, 1)$, $i = 1, 2$. Here, $(X_{1,1}, X_{2,1}, Y_1)$ is independent of $(X_{1,2}, X_{2,2}, Y_2)$. Using the additive property, we are able to aggregate the PID values derived from their separate decompositions, each of which is associated with a known ground truth, as determined by the MMI-PID, considering $Y_1$ and $Y_2$ as scalars. (ii) **Rotation**: Let $X_1 = H_1 R(\theta) Y$, where $H_1$ is a diagonal matrix with diagonal entries 3 and 1, and $R(\theta)$ is a $2 \times 2$ rotation matrix that rotates $Y$ at an angle $\theta$. When $\theta = 0$, $X_1$ has a higher gain for $Y_1$ and $X_2$ has higher gain for $Y_2$. When $\theta$ increases to $\pi/2$, $X_1$ and $X_2$ have equal gains for both $Y_1$ and $Y_2$ (barring a difference in sign). Since $(X_{1,1}, X_{2,1}, Y_1)$ is not independent of $(X_{1,2}, X_{2,2}, Y_2)$ for all $\theta$, we only know the ground truth at the endpoints.

**Results**. The results of examples (i) and (ii) are shown in Figure 3. We observe that Thin-PID achieves the best accuracy with the error $< 10^{-12}$, while the absolute error of Tilde-PID is $> 10^{-8}$.

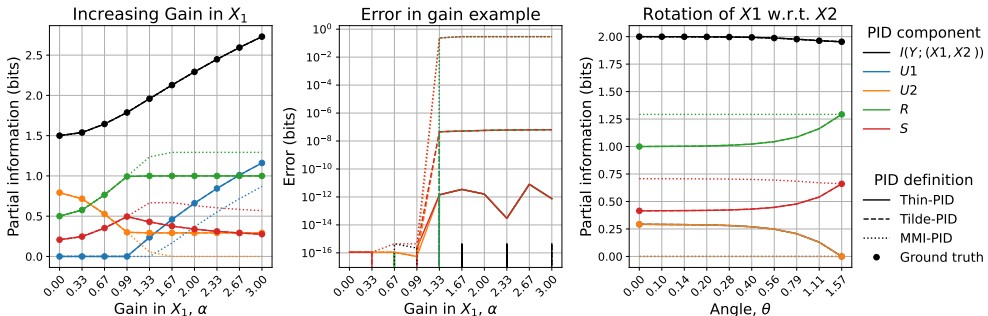

Figure 3: Left: PID values in Example (i); right: PID values in Example (ii); middle: absolute error between different GPID algorithms and the ground truth. Thin-PID achieves the best accuracy with $< 10^{-12}$ error, while the absolute error of Tilde-PID is $> 10^{-8}$.

Table 2: Non-Gaussian multi-dimensional examples: Tilde-PID is estimated from a sample covariance on $1e6$ realizations of transformed non-Gaussian variables. Flow-PID is estimated on the latent Gaussian representations by normalizing flows. Only Flow-PID agrees with the truth.

| Dim | (2, 2, 2) | | | | (10, 5, 2) | | | | (30, 10, 2) | | | | (100, 60, 2) | | | |
|---|---|---|---|---|---|---|---|---|---|---|---|---|---|---|---|---|
| PID | $R$ | $U_1$ | $U_2$ | $S$ | $R$ | $U_1$ | $U_2$ | $S$ | $R$ | $U_1$ | $U_2$ | $S$ | $R$ | $U_1$ | $U_2$ | $S$ |
| Tilde-PID | 0.18 | 0.29 | 0.76 | 0.02 | 0.84 | 0 | 1.19 | 0.16 | 1.09 | 0 | 0.88 | 0.19 | 1.48 | 0 | 1.97 | 0.13 |
| Flow-PID | **0.62** | **0.91** | **0.50** | **0.11** | **2.36** | **0.32** | **0.19** | **0.45** | **2.18** | **1.13** | **0** | **0.17** | **4.34** | **0.36** | **0** | **0.25** |
| Truth | 0.79 | 1.46 | 0.58 | 0.18 | 2.96 | 0.54 | 0.26 | 0.58 | 2.92 | 2.18 | 0 | 0.25 | 5.71 | 1.01 | 0 | 0.57 |

**Time analysis**: As discussed in Table 1, Thin-PID significantly improves the computational efficiency when the feature dimension is high. We report the execution time of GPID algorithms when the feature dimension increases in 2 cases: 1) both $d_{X_1}$ and $d_{X_2}$ increase; 2) $d_{X_1}$ increases from 100 to 1000 with fixed $d_{X_2} = 100$. Figure 4 shows that Thin-PID costs much less time than Tilde-PID with a speed of more than $10\times$ when $\min(d_{X_1}, d_{X_2}) > 100$.

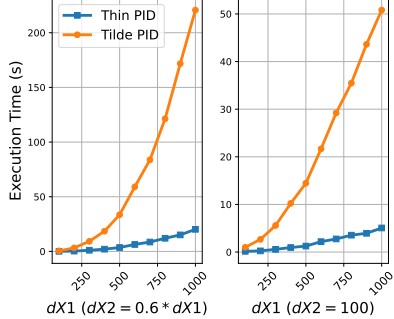

Figure 4: Time analysis: Thin-PID achieves $10\times$ speed of Tilde-PID.

## 5.2 Flow-PID on non-Gaussian examples

**Multivariate Gaussian with invertible nonlinear transformation**: Next we evaluate the Flow-PID when $p(x_i, y)$ is no longer Gaussian. We sample from the joint Gaussian vectors $(X_1, X_2, Y) \sim \mathcal{N}(0, \Sigma_{X_1 X_2 Y})$, and transform samples to $(\widetilde{X_1}, \widetilde{X_2}, \widetilde{Y})$ through absolutely invertible nonlinear function (more details in Appendix C.2). According to Theorem 4.1, the MI of $(\widetilde{X_1}, \widetilde{X_2}, \widetilde{Y})$ remains the same as that of $(X_1, X_2, Y)$, but they are no longer pairwise Gaussian after the transformation. For Tilde-PID, the degree of interactions are computed by directly estimating the covariance of non-Gaussian samples. The "exact" PID is obtained by feeding the Gaussian covariance $\Sigma_{X_1 X_2 Y}$ to GPID directly. From Table 2, Flow-PID aligns with the exact truth in relative PID values, whereas Tilde-PID fails with distorted nature and degree of interactions.

**Specialized interactions with discrete targets**: Although GPID is designed for continuous modalities and targets, we evaluate its generalization to cases with discrete targets. Three latent vectors $z_1, z_2, z_c \sim \mathcal{N}(0, \Sigma)$ are employed to quantify the information unique to $x_1$, $x_2$, and common to both, respectively. $[z_1, z_c]$ is transformed to high-dimensional $x_1$ using a fixed transformation $T_1$. Similarly, $[z_2, z_c]$ is transformed to $x_2$ via $T_2$. By assigning different weights to $[z_1, z_2, z_c]$, we create ten synthetic datasets with different types of specialized interactions. As indicated in Table 3, Flow-PID not only accurately assigns the prevalent type of interaction, but also provides better quantification of the degrees of specialized interactions compared to BATCH. A noteworthy observation is that the estimation of PID regarding synergy $S$ proves to be the most challenging and

Table 3: Specialized interactions with discrete labels: Flow-PID provides more accurate PID values than BATCH, especially on datasets with high synergistic interactions. The specialized interactions determined by PID estimators are highlighted in **bold**.

| Task | $\mathcal{D}_R$ | | | | $\mathcal{D}_{U_1}$ | | | | $\mathcal{D}_{U_2}$ | | | | $\mathcal{D}_S$ | | | |
|---|---|---|---|---|---|---|---|---|---|---|---|---|---|---|---|---|
| PID | $R$ | $U_1$ | $U_2$ | $S$ | $R$ | $U_1$ | $U_2$ | $S$ | $R$ | $U_1$ | $U_2$ | $S$ | $R$ | $U_1$ | $U_2$ | $S$ |
| BATCH | **0.29** | 0.02 | 0.02 | 0 | 0 | **0.30** | 0 | 0 | 0 | 0 | **0.30** | 0 | **0.11** | 0.02 | 0.02 | **0.15** |
| Flow-PID | **0.51** | 0 | 0 | 0 | 0 | **0.49** | 0 | 0 | 0 | 0 | **0.51** | 0 | 0.12 | 0 | 0 | **0.30** |
| Truth | 0.58 | 0 | 0 | 0 | 0 | 0.56 | 0 | 0 | 0 | 0 | 0.54 | 0 | 0 | 0 | 0 | 0.56 |

| Task | $y = f(z_1, z_2^*, z_c^*)$ | | | | $y = f(z_1, z_2, z_c^*)$ | | | | $y = f(z_1^*, z_2^*, z_c)$ | | | |
|---|---|---|---|---|---|---|---|---|---|---|---|---|
| PID | $R$ | $U_1$ | $U_2$ | $S$ | $R$ | $U_1$ | $U_2$ | $S$ | $R$ | $U_1$ | $U_2$ | $S$ |
| BATCH | **0.04** | **0.09** | 0 | **0.06** | **0.11** | 0.02 | 0.02 | **0.10** | **0.11** | 0 | 0.02 | **0.05** |
| Flow-PID | 0.06 | **0.17** | 0 | **0.12** | **0.20** | 0 | 0 | **0.23** | **0.10** | 0 | 0 | **0.08** |
| Truth | 0 | 0.25 | 0 | 0.25 | 0.18 | 0 | 0 | 0.36 | 0.22 | 0 | 0 | 0.22 |

| Task | $y = f(z_1^*, z_2^*, z_c^*)$ | | | | $y = f(z_2^*, z_c^*)$ | | | | $y = f(z_2^*, z_c)$ | | | |
|---|---|---|---|---|---|---|---|---|---|---|---|---|
| PID | $R$ | $U_1$ | $U_2$ | $S$ | $R$ | $U_1$ | $U_2$ | $S$ | $R$ | $U_1$ | $U_2$ | $S$ |
| BATCH | **0.06** | 0.01 | 0.01 | **0.06** | **0.07** | 0 | **0.06** | 0 | **0.19** | 0 | 0.06 | 0 |
| Flow-PID | **0.08** | 0 | 0 | **0.23** | **0.10** | 0 | **0.11** | 0 | **0.30** | 0 | **0.12** | 0 |
| Truth | 0.13 | 0 | 0 | 0.27 | 0.21 | 0 | 0.21 | 0.0 | 0.34 | 0 | 0.17 | 0 |

Table 4: Estimating PID on MultiBench datasets [30]. Flow-PID recognizes more modality interactions than CVX/BATCH, and effectively highlights dominant modalities.

| Datasets | AV-MNIST | | | | UR-FUNNY | | | | | | | | | | | |
|---|---|---|---|---|---|---|---|---|---|---|---|---|---|---|---|---|
| Modalities | Vision, Audio | | | | Vision, Audio | | | | Vision, Text | | | | Audio, Text | | | |
| PID | $R$ | $U_1$ | $U_2$ | $S$ | $R$ | $U_1$ | $U_2$ | $S$ | $R$ | $U_1$ | $U_2$ | $S$ | $R$ | $U_1$ | $U_2$ | $S$ |
| CVX/BATCH | 0.10 | **0.97** | 0.03 | 0.08 | 0.02 | 0.04 | 0 | **0.06** | 0.06 | 0 | 0.04 | **0.11** | 0.02 | 0 | **0.08** | 0.07 |
| Flow-PID | 0.12 | **0.53** | 0 | 0.09 | 0.19 | 0 | **0.98** | 0.28 | 0.04 | 0 | **1.19** | 0.03 | 0.03 | 0 | **1.22** | 0.12 |

| Datasets | MOSI | | | | MOSEI | | | | MUStARD | | | | | | | | | | | |
|---|---|---|---|---|---|---|---|---|---|---|---|---|---|---|---|---|---|---|---|---|
| Modalities | Vision, Audio | | | | Audio, Text | | | | Vision, Audio | | | | Vision, Text | | | | Audio, Text | | | |
| PID | $R$ | $U_1$ | $U_2$ | $S$ | $R$ | $U_1$ | $U_2$ | $S$ | $R$ | $U_1$ | $U_2$ | $S$ | $R$ | $U_1$ | $U_2$ | $S$ | $R$ | $U_1$ | $U_2$ | $S$ |
| CVX/BATCH | 0.03 | 0.17 | 0.16 | **0.76** | 0.22 | 0.04 | **0.09** | 0.13 | 0.14 | 0.01 | 0.01 | 0.2 | **0.14** | 0.02 | 0.01 | **0.34** | 0.14 | 0.01 | 0.01 | 0.37 |
| Flow-PID | **0.58** | 0 | **0.60** | 0.44 | **0.78** | 0 | **3.06** | 0.66 | **0.40** | 0 | **0.29** | 0.27 | **0.45** | 0 | **0.54** | 0.41 | **0.66** | 0 | **0.92** | 0.30 |

results in overestimated redundancy $R$ in BATCH. Conversely, Flow-PID mitigates this issue by achieving lower $R$ and higher $S$.

# 6 Real-world applications of PID

**Real-world multimodal benchmarks.** We use a collection of real-world multimodal datasets in MultiBench [30], which spans 10 diverse modalities (images, video, audio, text, time-series), 15 prediction tasks, and 5 research areas. These datasets are designed to test a combination of feature learning and arbitrarily complex interactions under different *multimodal fusion* models in the real world. For datasets with available modality features (images, text), we use an end-to-end Flow-PID estimator. For other modalities (audio, time-series), we first use pretrained encoders to obtain features before Flow-PID (full dataset and experimental settings are available in Appendix D).

**Results.** From Table 4, we observe that Flow-PID effectively highlights dominant modalities by assigning higher unique information to sources with stronger predictive contributions. Although the ground truth for real-world datasets is unknown (and may not be determined), this allows us to quantitatively assess which modalities are most informative for the task, offering deeper insight into modality importance beyond standard accuracy metrics. An interesting observation is that the total information ($R + U_1 + U_2 + S$) recognized by Flow-PID is much larger than BATCH. The total information in a dataset yields an upper bound on multimodal model performance. On many of these datasets, multimodal models achieve over 75% accuracy, while the total information is often under 0.5 for BATCH in Figure 5.

**Real-world datasets with task-driven and causally relevant interactions.** We conducted Flow-PID on 2 real-world datasets with expected interactions, which are causally relevant to the tasks, to demonstrate our method in additional application areas and with unexplored modalities. 1) We quantified the PID of predicting the breast cancer stage from protein expression and microRNA expression on the TCGA-BRCA dataset. Flow-PID identified *strong uniqueness* for the modality of microRNA expression as well as moderate amounts of redundancy and synergy. These results are

also in line with modern research, which suggests microRNA changes as a direct result of cancer progression. 2) The results on VQA (Visual Question Answering) show the expected *high synergy* since the image and question complement each other to predict the answer. The detailed experimental results are shown in Table 5.

Table 5: Flow-PID on real datasets with task-driven interactions.

| Dataset | dim-$X_1$ | dim-$X_2$ | $R$ | $U_1$ | $U_2$ | $S$ | Expected interaction |
|---------|-----------|-----------|-----|-------|-------|-----|----------------------|
| TCGA | 217 | 1881 | 0.41 | 0.0 | **1.07** | 0.34 | $U_2$ |
| VQA2.0 | 768 | 1000 | 0.22 | 0.26 | 0.0 | **0.76** | $S$ |

Table 6: Model selection performance on new datasets $\mathcal{D}$ compared to the best-performing model.

| | Synthetic | AV-MNIST | ENRICO | UR-FUNNY | MOSI | MUStARD |
|---|-----------|----------|--------|----------|------|---------|
| Flow-PID | 99.76% | 100% | 100% | 96.72% | 99.67% | 98.19% |
| BATCH | 99.91% | 99.85% | 100% | 98.58% | 99.35% | 95.15% |

**Model Selection.** After conducting evaluations on diverse multimodal datasets, we are interested in whether GPID is beneficial in selecting the most appropriate model capable of addressing the requisite interactions for a dataset. Given a new dataset $\mathcal{D}$, we measure the difference in normalized PID values between $\mathcal{D}$ and $\mathcal{D}'$ among a suite of 10 pretrained synthetic datasets with different types of specialized interactions. For each $\mathcal{D}'$, we pretrain 8 different multimodal fusions, rank the similarities to the unseen dataset, $s(\mathcal{D}, \mathcal{D}') = \sum_{I \in \{R, U_1, U_2, S\}} |I_{\mathcal{D}} - I_{\mathcal{D}'}|$, and recommend the top-3 models on the most similar dataset $\mathcal{D}^*$. Table 6 indicates that the selected models achieve more than 96% of the accuracy of the best-performing model. UR-FUNNY records the comparatively lower accuracy, likely due to the significantly higher amount of unique information in the text modality compared to vision and audio.

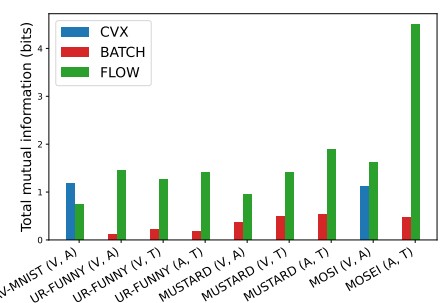

Figure 5: Total mutual information determined by Flow-PID and BATCH/CVX estimators.

# 7 Conclusion

In this paper, we aim to develop a new and efficient PID framework for continuous and high-dimensional modalities, of which PID estimates could be inaccurate and burdensome. We first identified that PID is easier to solve in latent Gaussian distributions without loss of optimality, and proposed a new GPID algorithm that significantly enhances the computational efficiency compared to the state-of-the-art algorithms. Secondly, we develop a latent Gaussian encoder via normalizing flows to generalize GPID algorithms to non-Gaussian cases. Through comprehensive experiments, we demonstrated that our proposed method provides more accurate and efficient PID estimates than existing baselines, and showed the utility in diverse multimodal datasets and applications by dataset quantification and model selection.

**Limitations**: 1) The latent Gaussian encoders only approximate the marginal distributions through invertible bijective mappings, which introduce bias when the divergence between approximated Gaussian distributions and true underlying marginals is large. 2) Although Thin-PID is exact in GPID cases, the accuracy of Flow-PID largely depends on the performance of latent Gaussian encoders, and the optimization error could increase with more intricate underlying features or fewer data samples. 3) It is challenging to rigorously justify the quantification on real-world datasets, as the generation of multimodal data is unknown.

**Future work** can leverage Flow-PID to expand datasets with specific objectives, enhance multi-task representation learning within the context of higher-dimensional data and continuous targets, and explore the fine-tuning or pretraining of a large model under the guidance of Flow-PID.

## Acknowledgments

The work of Wenyuan Zhao and Chao Tian is partly supported by NSF via grant DMS-2312173. We also acknowledge Nvidia for their GPU support.

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

# Appendix

## A  The Gaussian PID theory and algorithm

Information theory quantifies how much information one variable $X$ offers about another variable $Y$, which is formulated by *Shannon's mutual information* [9], represented as $I(X;Y)$. It reflects the decrease in entropy from $H(Y)$ to $H(Y|X)$ given $X$ as input. However, extending mutual information (MI) directly to three or more variables presents notable challenges. Specifically, the three-way MI $I(X_1; X_2; Y)$ can be both negative and positive, leading to considerable difficulty in its interpretation when quantifying interactions between multiple variables.

Partial Information Decomposition (PID) [55] was introduced as a framework to extend information theory to multiple variables. It decomposes the total information that two variables offer about a task $I(X_1, X_2; Y)$ into four components: redundancy $R$ shared between $X_1$ and $X_2$, unique information $U_1$ specific to $X_1$ and $U_2$ to $X_2$, and synergy $S$. These components must collectively fulfill the following consistency equations:

$$R + U_1 = I(X_1; Y), \tag{18}$$
$$R + U_2 = I(X_2; Y), \tag{19}$$
$$U_1 + S = I(X_1; Y | X_2), \tag{20}$$
$$U_2 + S = I(X_2; Y | X_1), \tag{21}$$
$$R - S = I(X_1; X_2; Y). \tag{22}$$

A definition of $R$ was first proposed by Williams and Beer [55] and subsequently improved by Bertschinger et al. [5], Griffith and Koch [17], which gives the PID definition we adopt in this work.

**Definition A.1** (PID [27]). The redundant, unique, and synergistic information are given by

$$R = \max_{q \in \Delta_p} I_q(X_1; X_2; Y), \tag{23}$$

$$U_1 = \min_{q \in \Delta_p} I_q(X_1; Y | X_2), \quad U_2 = \min_{q \in \Delta_p} I_q(X_2; Y | X_1), \tag{24}$$

$$S = I_p(X_1, X_2; Y) - \min_{q \in \Delta_p} I_q(X_1, X_2; Y), \tag{25}$$

where $\Delta_p := \{q \in \Delta : q(x_i, y) = p(x_i, y), \ \forall y \in \mathcal{Y}, x_i \in \mathcal{X}_i, i \in [2]\}$, and $I_q$ is the mutual information (MI) over the joint distribution $q(x_1, x_2, y)$. Note that $\Delta_p$ only preserves the marginals $p(x_1, y)$ and $p(x_2, y)$, but not necessarily the joint distribution $p(x_1, x_2, y)$.

The definition of PID enjoys two properties:

1. **Non-negativity**: all the four decomposed components $(R, U_1, U_2, S)$ are non-negative.
2. **Additivity**: For two independent subsystems $(Y_1, X_{1,1}, X_{2,1})$ and $(Y_2, X_{1,2}, X_{2,2})$, we have $U_1(Y : X_1 \setminus X_2) = U_1(Y_1 : X_{1,1} \setminus X_{2,1}) + U_1(Y_2 : X_{1,2} \setminus X_{2,2})$. This implies that the PID of an isolated system should not depend on another isolated system.

The fundamental challenge of PID is estimating information-theoretic measures when the size and dimensionality of the datasets are large [43]. Optimizing over the pointwise MI quantities can only be obtained when the features are pointwise discrete or low-dimensional, and the number of optimization variables is exponential in the number of neurons [53]. Our first key insight is that the measurement of MI has a closed-form analysis when the pairwise distributions are multivariate Gaussians, and we refer to this problem as Gaussian PID (GPID).

**Definition A.2** (GPID). Let $\Delta^{\mathrm{G}}$ be the set of joint distributions, where $p(x_1, y)$ and $p(x_2, y)$ are pairwise Gaussian. The redundant, unique, and synergistic information are given by

$$R = \max_{q \in \Delta_p^{\mathrm{G}}} I_q(X_1; X_2; Y), \tag{26}$$

$$U_1 = \min_{q \in \Delta_p^{\mathrm{G}}} I_q(X_1; Y | X_2), \quad U_2 = \min_{q \in \Delta_p^{\mathrm{G}}} I_q(X_2; Y | X_1), \tag{27}$$

$$S = I_p(X_1, X_2; Y) - \min_{q \in \Delta_p^{\mathrm{G}}} I_q(X_1, X_2; Y), \tag{28}$$

where

$$\Delta_p^{\mathrm{G}} := \{q \in \Delta^{\mathrm{G}} : q(x_i, y) = p(x_i, y), \ \forall y \in \mathcal{Y}, \ x_i \in \mathcal{X}_i, \ i \in [2]\}. \tag{29}$$

**Definition A.3** ($\sim_G$-PID [52]). Let $p(x_1, x_2, y)$ be a joint Gaussian distribution. The redundant, unique, and synergistic information are given by

$$R = \max_{q \in \Delta_p} I_q(X_1; X_2; Y), \tag{30}$$

$$U_1 = \min_{q \in \Delta_p} I_q(X_1; Y | X_2), \quad U_2 = \min_{q \in \Delta_p} I_q(X_2; Y | X_1), \tag{31}$$

$$S = I_p(X_1, X_2; Y) - \min_{q \in \Delta_p} I_q(X_1, X_2; Y), \tag{32}$$

where

$$\Delta_p := \{q \text{ is jointly Gaussian} : q(x_i, y) = p(x_i, y), \ \forall y \in \mathcal{Y}, \ x_i \in \mathcal{X}_i, \ i \in [2]\}. \tag{33}$$

**Connections between PID definitions**: GPID is exactly the PID problem when the pairwise marginals $p(x_1, y)$ and $p(x_2, y)$ are known to be Gaussians. If the optimal $q(x_1, x_2, y)$ in PID is Gaussian for some $p(x_1, x_2, y)$, then $\sim_G$-PID is identical to PID for that $p(x_1, x_2, y)$. Venkatesh et al. [52] introduced $\sim_G$-PID by directly placing the additional constraint that $q(x_1, x_2, y)$ is Gaussian. In Section 3.1, we show that the joint Gaussian solution in $\sim_G$-PID is also optimal in GPID, but this was left as an open question in [52].

**Broadcast channel interpretation of GPID**: Let two Gaussian marginals in GPID have covariance $\Sigma_{X_1Y}$ and $\Sigma_{X_2Y}$, respectively. We can interpret GPID in the following two-user Gaussian broadcast channel:

$$X_1 = H_1 Y + n_1, \tag{34}$$
$$X_2 = H_2 Y + n_2, \tag{35}$$

where $H_1, H_2, n_1, n_2$ can be estimated from $p(x_1, y)$ and $p(x_2, y)$:

$$[X_1^\top, Y^\top]^\top \sim \mathcal{N}(\mu_1, \Sigma_{X_1Y}), \tag{36}$$

$$[X_2^\top, Y^\top]^\top \sim \mathcal{N}(\mu_2, \Sigma_{X_2Y}), \tag{37}$$

$$H_1 = \Sigma_{X_1Y}^{\text{off}}, \tag{38}$$

$$H_2 = \Sigma_{X_2Y}^{\text{off}}, \tag{39}$$

$$\Sigma_{n_1} = \Sigma_{X_1} - H_1 \Sigma_Y H_1^\top, \tag{40}$$

$$\Sigma_{n_2} = \Sigma_{X_2} - H_2 \Sigma_Y H_2^\top. \tag{41}$$

**Full algorithm of Thin-PID**: We derived the objective and the projected gradient descent method in Theorem 3.3 and Proposition 3.4, respectively. The complete algorithm of Thin-PID is given as follows.

---
**Algorithm 1** Thin-PID algorithm.

---
**Require:** Channel matrix $H = [H_1^\top, H_2^\top]^\top$, covariance $\Sigma_Y$.

    **Initialize:** $\Sigma_{n_1n_2}^{\text{off}(0)} = H_1 H_2^+$, learning rate $\eta^{(0)}$, $\alpha = 0.999$, $\beta = 0.9$

    **while** not converged **do**

        Compute $\nabla \mathcal{L}\left(\Sigma_{n_1n_2}^{\text{off}(j)}\right) = \left[(H\Sigma_Y H^\top + \Sigma_{n_1n_2})^{-1} - \Sigma_{n_1n_2}^{-1}\right]_{\text{up-off}}$ from Eq. (12-13)

        Update $\Sigma_{n_1n_2}^{\text{off}}$ using RProp [42]: $\Sigma_{n_1n_2}^{\text{off}(j+1)} \leftarrow \Sigma_{n_1n_2}^{\text{off}(j)} - \alpha^j \eta^{(j)} \odot \nabla \mathcal{L}\left(\Sigma_{n_1n_2}^{\text{off}(j)}\right)$

        $\text{SVD}(\Sigma_{n_1n_2}^{\text{off}(j+1)}) = U\text{diag}(\lambda_i)V^\top$ from Eq. (14)

        $\text{Proj}(\Sigma_{n_1n_2}^{\text{off}(j+1)}) = U\text{diag}\left(\min\left(\max(\lambda_i, 0), 1\right)\right)V^\top$ from Eq. (15)

        Update $\eta^{(j+1)} = \eta^{(j)} \odot \beta^{-\psi(\Sigma_{n_1n_2}^{\text{off}(j+1)}) \odot \psi(\Sigma_{n_1n_2}^{\text{off}(j)})}$, $\psi(\Sigma_{n_1n_2}^{\text{off}(j)}) := \text{Sgn}(\nabla\mathcal{L}(\Sigma_{n_1n_2}^{\text{off}(j)}))$

    **end while**

    **return** $\Sigma_{n_1n_2}^{\text{off}}$

---

## A.1 Proof of Lemma 3.2

For any random vectors $X_1, X_2, Y$ that follow the distribution $q(x_1, x_2, y)$,

$$h_q(Y|X_1, X_2) = h_q\left(Y - \mathbb{E}(Y|X_1, X_2)|X_1, X_2\right) \tag{42}$$

$$\leq h_q\left(Y - \mathbb{E}(Y|X_1, X_2)\right) \tag{43}$$

$$\leq h_q\left(\mathcal{N}(0, \Sigma_{Y-\mathbb{E}(Y|X_1, X_2)})\right) \tag{44}$$

$$= h_{\hat{q}}\left(\hat{Y} - \mathbb{E}(\hat{Y}|\hat{X}_1, \hat{X}_2)\right) \tag{45}$$

$$= h_{\hat{q}}\left(\hat{Y}|\hat{X}_1, \hat{X}_2\right) \tag{46}$$

where for clarity we also use $\hat{X}_1, \hat{X}_2, \hat{Y}$ to denote the random variables that follow the distribution $\hat{q}(x_1, x_2, y)$; the second inequality is because Gaussian distributions maximize the differential entropy with the same second moment. $h\left(\mathcal{N}(0, \Sigma_{Y-\mathbb{E}(Y|X_1, X_2)})\right)$ denotes the differential entropy of Gaussian vector with zero mean and covariance the same as $Y - \mathbb{E}(Y|X_1, X_2)$, and the last two equalities are because of the joint Gaussian distribution.

## A.2 Proof of Theorem 3.3

The optimization problem we need to solve in Gaussian PID is

$$S = I_p(X_1, X_2; Y) - \min_{q \in \Delta_p^{\mathrm{G}}} I_q(X_1, X_2; Y), \tag{47}$$

where

$$\Delta_p^{\mathrm{G}} := \{q \in \Delta^{\mathrm{G}} : q \text{ is jointly Gaussian, } q(x_i, y) = p(x_i, y), \ \forall y \in \mathcal{Y}, \ x_i \in \mathcal{X}_i, \ i \in [2]\}. \tag{48}$$

The random variables in the Gaussian PID interpreted by a Gaussian broadcast channel can be written as

$$\begin{bmatrix} X_1 \\ X_2 \end{bmatrix} = \begin{bmatrix} H_1 \\ H_2 \end{bmatrix} Y + \begin{bmatrix} n_1 \\ n_2 \end{bmatrix}, \tag{49}$$

where $Y \sim \mathcal{N}(0, \Sigma_Y)$, $n_1 \sim \mathcal{N}(0, \Sigma_{n_1})$, $n_2 \sim \mathcal{N}(0, \Sigma_{n_2})$, and $H = [H_1^\top, H_2^\top]^\top$ is the channel matrix which can be directly estimated from Gaussian marginals $p(x_1, y)$ and $p(x_2, y)$.

We firstly consider the objective function $I_q(X_1, X_2; Y)$. The differential entropy of a Gaussian random vector $X \sim \mathcal{N}(\mu_X, \Sigma_X)$ is given by

$$h(X) = \frac{n}{2} \log(2\pi) + \frac{1}{2} \log |\Sigma_X| + \frac{1}{2}n, \tag{50}$$

where $n$ is the dimension of vector $X$. Therefore, the MI between two random vectors $X \sim \mathcal{N}(\mu_X, \Sigma_X)$ and $Y \sim \mathcal{N}(\mu_Y, \Sigma_Y)$ is given by

$$I(X; Y) = h(X) - h(X|Y) \tag{51}$$

$$= \frac{1}{2} \log \frac{|\Sigma_X|}{|\Sigma_{X|Y}|}. \tag{52}$$

Therefore, the objective we need to optimize in synergistic information $S$ is

$$I_q(X_1, X_2; Y) = \frac{1}{2} \log \frac{|\Sigma_{X_1 X_2}|}{|\Sigma_{X_1 X_2|Y}|} \tag{53}$$

$$= \frac{1}{2} \log \frac{|H\Sigma_Y H^\top + \Sigma_{n_1 n_2}|}{|\Sigma_{n_1 n_2}|}, \tag{54}$$

where

$$\Sigma_{n_1 n_2} := \begin{bmatrix} \Sigma_{n_1} & \Sigma_{n_1 n_2}^{\mathrm{off}} \\ \Sigma_{n_2 n_1}^{\mathrm{off}} & \Sigma_{n_2} \end{bmatrix}. \tag{55}$$

The second equality is because $\Sigma_{X_1 X_2 | Y} = \Sigma_{n_1 n_2}$ and $\Sigma_{X_1 X_2} = H \Sigma_Y H^\top + \Sigma_{n_1 n_2}$ using properties of multivariate Gaussian distributions.

Next, we consider the constraints on Gaussian marginals $q(x_1, y) = p(x_1, y)$, $q(x_2, y) = p(x_2, y)$. Note that $p(x_1, y)$ and $p(x_2, y)$ are already preserved by $\Sigma_{n_1}$ and $\Sigma_{n_2}$. Without loss of generality, we can assume $\Sigma_{X_1 | Y} = \Sigma_{n_1} = I_{d_{X_1}}$ and $\Sigma_{X_2 | Y} = \Sigma_{n_2} = I_{d_{X_2}}$ since we can always perform receiver side linear transformations to individually whiten the $X_1$ and $X_2$ channels. Therefore, the only optimization variable in $I_q(X_1, X_2; Y)$ is $\Sigma_{n_1 n_2}^{\text{off}}$.

Therefore, the optimization problem can be recast as

$$
\begin{aligned}
\text{minimize: } & \mathcal{L}\left(\Sigma_{n_1 n_2}^{\text{off}}\right) = \log \frac{\left| H \Sigma_Y H^\top + \Sigma_{n_1 n_2} \right|}{\left| \Sigma_{n_1 n_2} \right|}, \\
\text{subject to: } & \Sigma_{n_1} = \Sigma_{n_2} = I, \\
& \Sigma_{n_1 n_2} \succeq 0.
\end{aligned}
\tag{56}
$$

## A.3 Proof of Proposition 3.4

**Gradient.** Let $G := H \Sigma_Y H^\top + \Sigma_{n_1 n_2}$. Assuming that $G$ is positive definite, then we have

$$
\nabla_G \log \det(G) = G^{-1}.
\tag{57}
$$

Therefore, the gradient of the unconstrained objective function in Equation (11) with respect to $\Sigma_{n_1 n_2}$ is given by

$$
\nabla_{\Sigma_{n_1 n_2}} \log \frac{|G|}{|\Sigma_{n_1 n_2}|} = G^{-1} - \Sigma_{n_1 n_2}^{-1}.
\tag{58}
$$

Using the block matrix formulas, the gradient with respect to $\Sigma_{n_1 n_2}^{\text{off}}$ is thus given by

$$
\nabla \mathcal{L}\left(\Sigma_{n_1 n_2}^{\text{off}}\right) = \left[ G^{-1} - \Sigma_{n_1 n_2}^{-1} \right]_{\text{up-off}},
\tag{59}
$$

where the subscript "up-off" denotes the *upper off-diagonal* block matrix.

We firstly compute the inverse of $G := H \Sigma_Y H^\top + \Sigma_{n_1 n_2}$. Define the block matrices

$$
H \Sigma_Y H^\top + \Sigma_{n_1 n_2} := \begin{bmatrix} G_{1,1} & G_{1,2} \\ G_{1,2}^\top & G_{2,2} \end{bmatrix}, \ \Sigma_{n_1 n_2} := \begin{bmatrix} \Sigma_{n_1} & \Sigma_{n_1 n_2}^{\text{off}} \\ \Sigma_{n_2 n_1}^{\text{off}} & \Sigma_{n_2} \end{bmatrix},
\tag{60}
$$

where

$$
G_{1,1} = H_1 \Sigma_Y H_1^\top + \Sigma_{n_1},
\tag{61}
$$

$$
G_{2,2} = H_2 \Sigma_Y H_2^\top + \Sigma_{n_2},
\tag{62}
$$

$$
G_{1,2} = H_1 \Sigma_Y H_2^\top + \Sigma_{n_1 n_2}^{\text{off}}.
\tag{63}
$$

Using the block matrix inverse formulas, the upper off-diagonal of $G^{-1}$ is given by

$$
-G_{1,1}^{-1} G_{1,2} \left( G_{2,2} - G_{1,2}^\top G_{1,1}^{-1} G_{1,2} \right)^{-1}.
\tag{64}
$$

Similarly, the upper of-diagonal of $\Sigma_{n_1 n_2}^{-1}$ is given by

$$
-\Sigma_{n_1 n_2}^{\text{off}} \left( I - \Sigma_{n_1 n_2}^{\text{off} \top} \Sigma_{n_1 n_2}^{\text{off}} \right)^{-1},
\tag{65}
$$

if $\Sigma_{n_1}$ and $\Sigma_{n_2}$ are identity matrices without loss of generality.

Therefore, the gradient of the unconstrained objective is given by

$$
\nabla \mathcal{L}\left(\Sigma_{n_1 n_2}^{\text{off}}\right) = -G_{1,1}^{-1} G_{1,2} \left( G_{2,2} - G_{1,2}^\top G_{1,1}^{-1} G_{1,2} \right)^{-1} + \Sigma_{n_1 n_2}^{\text{off}} \left( I - \Sigma_{n_1 n_2}^{\text{off} \top} \Sigma_{n_1 n_2}^{\text{off}} \right)^{-1}.
\tag{66}
$$

**Projection operator.** The optimization variable $\Sigma^{\mathrm{off}}_{n_1 n_2}$ is an off-diagonal block of $\Sigma_{n_1 n_2}$, which is the matrix constrained by

$$\Sigma_{n_1 n_2} := \begin{bmatrix} I & \Sigma^{\mathrm{off}}_{n_1 n_2} \\ \Sigma^{\mathrm{off}}_{n_2 n_1} & I \end{bmatrix}. \tag{67}$$

Note that $\Sigma_{n_1}$ and $\Sigma_{n_2}$ can always be whitened by performing receiver-side linear transformations after we estimate $H_1, H_2, \Sigma_Y$ from the data. Therefore, the constraint on the projection we need to consider is

$$\begin{bmatrix} I & \Sigma^{\mathrm{off}}_{n_1 n_2} \\ \Sigma^{\mathrm{off}}_{n_2 n_1} & I \end{bmatrix} \succeq 0. \tag{68}$$

By the Schur complement conditions for positive definiteness,

$$\Sigma_{n_1 n_2} \succeq 0 \iff \left\| \Sigma^{\mathrm{off}}_{n_1 n_2} \right\|_2 \leq 1. \tag{69}$$

Therefore, we only need to show that the projection of $\Sigma^{\mathrm{off}}_{n_1 n_2}$ onto the spectral norm ball $\left\{ \Sigma^{\mathrm{off}}_{n_1 n_2} : \|\Sigma^{\mathrm{off}}_{n_1 n_2}\|_2 \leq 1 \right\}$ is achieved by shrinking the singular values of $\Sigma^{\mathrm{off}}_{n_1 n_2}$ via

$$\mathrm{Proj}(\Sigma^{\mathrm{off}}_{n_1 n_2}) = U \mathrm{diag}\left(\min\left(\max(\lambda_i, 0), 1\right)\right) V^\top, \tag{70}$$

where $\Sigma^{\mathrm{off}}_{n_1 n_2} = U \Lambda V^\top$ is the SVD of $\Sigma^{\mathrm{off}}_{n_1 n_2}$.

To find the projection onto the spectral norm ball, we want to solve

$$\min_{\Sigma} \|\Sigma - \Sigma^{\mathrm{off}}_{n_1 n_2}\|^2_F, \tag{71}$$

$$\text{s.t. } \|\Sigma\|_2 \leq 1. \tag{72}$$

Let $\Sigma^{\mathrm{off}}_{n_1 n_2} = U \Lambda V^\top$ with $\Lambda := \mathrm{diag}(\lambda_i)$. We apply the same decomposition to $\Sigma = U \bar{\Lambda} V^\top$, where $\bar{\Lambda}$ is not necessarily diagonal. However, we can always set the off-diagonal blocks to zeros without increasing the Frobenius norm.

Write $\Sigma = D + O$, where $D$ is the diagonal matrix of $\Sigma$ and $O$ is the pure off-diagonal matrix. Since the Frobenius inner product is the usual Euclidean one on entries, the diagonal and off-diagonal subspaces are orthogonal. Hence, we have

$$\left\|\Sigma - \Sigma^{\mathrm{off}}_{n_1 n_2}\right\|^2_F = \left\|D - \Sigma^{\mathrm{off}}_{n_1 n_2}\right\|^2_F + \|O\|^2_F \geq \left\|D - \Sigma^{\mathrm{off}}_{n_1 n_2}\right\|^2_F. \tag{73}$$

For the feasibility of the spectral norm ball, it is obvious because

$$\|\mathrm{diag}(\Sigma)\|_2 = \max_i |\sigma_{ii}| \leq \|\Sigma\|_2. \tag{74}$$

Now it is feasible to restrict $\bar{\Lambda}$ to be a diagonal matrix. Because the Frobenius norm is unitary-invariant, we can simplify:

$$\|\Sigma - \Sigma^{\mathrm{off}}_{n_1 n_2}\|^2_F = \|\bar{\Lambda} - \Lambda\|^2_F = \sum_{i=1}^r \left(\bar{\lambda}_i - \lambda_i\right)^2. \tag{75}$$

Thus, we only need to solve:

$$\min_{\{\bar{\lambda}_i\}} \sum_{i=1}^r \left(\bar{\lambda}_i - \lambda_i\right)^2 \quad \text{s.t. } \max_i |\bar{\lambda}_i| \leq 1. \tag{76}$$

Note that the SVD of real matrices always gives nonnegative eigenvalues. Therefore, $\lambda_i$ and $\bar{\lambda}_i$ should be non-negative, and the optimal solution is given by

$$\bar{\lambda}_i = \min(\max(\lambda_i, 0), 1). \tag{77}$$

Then the projection operator is given by

$$\mathrm{Proj}(\Sigma^{\mathrm{off}}_{n_1 n_2}) \leftarrow U \mathrm{diag}\left(\bar{\lambda}_i\right) V^\top. \tag{78}$$

## A.4 Computational complexity

In this section, we discuss the computational complexity of different GPID algorithms. The state-of-the-art GPID algorithm is Tilde-PID [52], which is shown to be faster than other older baselines (MMI-PID [3], $\delta$-PID [53]). We identify the difference between our work and Tilde-PID as follows:

(i) We compute the PID of a different objective function. Although the optimized variable is the same, $\Sigma_{n_1 n_2}$, the computation of the gradient in each iteration is significantly different. For Thin-PID, $G_{1,1}^{-1}$ in Eq. (12) is a constant and does not need to be computed in each iteration. The other inverses of the matrix in each iteration can be computed by solving a set of linear equations with variables $\min(d_{X_1}, d_{X_2})$, while Tilde-PID requires solving linear equations with dominant variables of size $d_{X_1} + d_{X_2}$.

(ii) We use a different projection operator on $\Sigma_{n_1 n_2}$. Although the constraint is $\Sigma_{n_1 n_2} \succeq 0$, we only work on the upper off-diagonal $\Sigma_{n_1 n_2}^{\text{off}}$ since the diagonal blocks are identity matrices. The Thin-PID requires SVD in $\Sigma_{n_1 n_2}^{\text{off}}$, which has $O\big(\min(d_{X_1}, d_{X_2})^3\big)$ complexity. However, Tilde-PID requires the eigenvalue decomposition in $\Sigma_{n_1 n_2}$ of size $(d_{X_1} + d_{X_2}) \times (d_{X_1} + d_{X_2})$, which has $O((d_{X_1} + d_{X_2})^3)$ complexity, and additional SVD in the projector with $O(\max(d_{X_1}, d_{X_2})^3)$ complexity.

Table A.1: Complexity analysis of different GPID algorithms. The complexities of ED and SVD are cubic in the values shown in the table. Thin-PID achieves better complexity on any scale of computation.

|  | ED | SVD | Lin-Eqn Solve | Mul |
|---|---|---|---|---|
| Thin-PID | – | $\min(d_{X_1}, d_{X_2})$ | $2 * \min(d_{X_1}, d_{X_2})$ | $4 * \min(d_{X_1}, d_{X_2})$ |
| Tilde-PID | $d_{X_1} + d_{X_2}$ | $\max(d_{X_1}, d_{X_2})$ | $2 * (d_{X_1} + d_{X_2})$ | $8 * (d_{X_1} + d_{X_2})$ |

# B Latent Gaussian encoders and normalizing flows

When the marginal distributions $p(x_1, y)$ and $p(x_2, y)$ are not Gaussian, we learn a feature encoder transforming $(X_1, X_2, Y)$ into a latent space such that they are well-approximated by Gaussian marginal distributions, then using Thin-PID to compute PID values in the GPID problem.

**Information-preserving encoder**: Let $\hat{X}_1 = f_1(X_1)$, $\hat{X}_2 = f_2(X_2)$, $\hat{Y} = f_Y(Y)$ be three transformations defined by three neural networks, respectively. Ideally, the transformations should satisfy the following conditions:

1. Transformations are invertible, so that the MI does not change.
2. $p(\hat{x}_1, \hat{y})$ and $p(\hat{x}_2, \hat{y})$ are well approximated by Gaussian distributions.

**Flow-PID**: This transformation can be performed by normalizing flows [39], whose invertible bijections preserve information while bringing the joint distributions closer to Gaussians [7]. We established the theory of invariant total MI and PID under bijective mappings in Theorem 4.1 and Corollary 4.2. Our goal is to learn a Cartesian product of normalizing flows $f_1 \times f_2 \times f_Y$ that preserves the total mutual information as $I_{\hat{p}}(f_1(X_1), f_2(X_2); f_Y(Y)) = I_p(X_1, X_2; Y)$. For the constraint set on $\Delta_p^{\text{G}}$, we proposed the regularization with the Gaussian marginal loss in Corollary 4.3 and Proposition 4.4.

## B.1 Proof of Theorem 4.1

**Lemma B.1** ([7]). *Let $\xi : \Omega \to \mathbb{R}^{n'}$ be an absolutely continuous random vector, and let $g : \mathbb{R}^{n'} \to \mathbb{R}^n$ be an injective piecewise-smooth mapping with Jacobian $J$, satisfying $n \geq n'$ and $\det(J^\top J) \neq 0$ almost everywhere. Let PDFs $p_\xi$ and $p_{\xi|\eta}$ exist. Then,*

$$I(\xi; \eta) = I(g(\xi); \eta). \tag{79}$$

By Lemma B.1, we only need to show $I(f_1(X_1), f_2(X_2); Y) = I(X_1, X_2; Y)$. Let $g(X_1, X_2) = [f_1(X_1), f_2(X_2)]$ be the concatenation of $f_1(X_1)$ and $f_2(X_2)$. It is obvious that concatenation is also bijective and invertible. Therefore, we have

$$I(f_1(X_1), f_2(X_2); Y) = I(g(X_1, X_2); Y) = I(X_1, X_2; Y), \tag{80}$$

where the second equality is because $g(X_1, X_2)$ is invertible and bijective.

---

**Algorithm 2** Flow-PID algorithm

---
**Require:** Multimodal dataset $\mathbf{X}_1 \in \mathcal{X}_1^n$, $\mathbf{X}_2 \in \mathcal{X}_2^n$, $\mathbf{Y} \in \mathcal{Y}^n$.
    Initialize Cartesian flow networks $f_1 \times f_2 \times f_Y$.
    **while** not converged **do**
        **for** sampled batch $\mathbf{X}_1 \in \mathcal{X}_1^m$, $\mathbf{X}_2 \in \mathcal{X}_2^m$, $\mathbf{Y} \in \mathcal{Y}^m$ **do**
            Transform latent feature: $\hat{\mathbf{X}}_1 = f_1(\mathbf{X}_1)$, $\hat{\mathbf{X}}_2 = f_2(\mathbf{X}_2)$, $\hat{\mathbf{Y}} = f_Y(\mathbf{Y})$.
            Compute the marginal loss of $(\hat{\mathbf{X}}_1, \hat{\mathbf{Y}})$ and $(\hat{\mathbf{X}}_2, \hat{\mathbf{Y}})$ using Eq. (16).
            Compute the sum of two marginal losses using Eq. (17).
            Perform a gradient step on the loss.
        **end for**
    **end while**
    Calculate $H_1, H_2, \Sigma_Y$ from Eq. (10) using $(\hat{\mathbf{X}}_1, \hat{\mathbf{X}}_2, \hat{\mathbf{Y}})$
    Perform Thin-PID from Eq. (11-15) using $H_1, H_2, \Sigma_Y$
    **return** PID values: $R, U_1, U_2, S$

---

## B.2 Proof of Corollary 4.2

Let $\hat{X}_1 = f_1(X_1)$, $\hat{X}_2 = f_2(X_2)$, and $\hat{Y} = f_Y(Y)$, where $f_1, f_2, f_Y$ are invertible bijective mappings. Define the set of distributions

$$\Delta_{\hat{p}} = \left\{ \hat{q} : \hat{q}(\hat{x}_1, \hat{y}) = p(f_1^{-1}(\hat{x}_1), f_Y^{-1}(\hat{y})), \ \hat{q}(\hat{x}_2, \hat{y}) = p(f_2^{-1}(\hat{x}_2), f_Y^{-1}(\hat{y})) \right\}.$$

Since $f_1$, $f_2$, and $f_Y$ are bijective mappings, this set is well-defined. Moreover, if $(X_1, X_2, Y)$ are jointly distributed according to $p$, then $(\hat{X}_1, \hat{X}_2, \hat{Y})$ are jointly distributed according to

$$\hat{p}(\hat{x}_1, \hat{x}_2, \hat{y}) = p(f_1^{-1}(\hat{x}_1), f_2^{-1}(\hat{x}_2), f_Y^{-1}(\hat{y})).$$

We now show that there is a bijection between the sets $\Delta_p$ and $\Delta_{\hat{p}}$. Given any $q \in \Delta_p$, define $\hat{q} \in \Delta_{\hat{p}}$ by

$$\hat{q}(\hat{x}_1, \hat{x}_2, \hat{y}) = q(f_1^{-1}(\hat{x}_1), f_2^{-1}(\hat{x}_2), f_Y^{-1}(\hat{y})).$$

It is straightforward to verify that $\hat{q}$ satisfies the required marginal constraints in $\Delta_{\hat{p}}$, as the marginals transform correctly under invertible mappings. Conversely, given any $\hat{q} \in \Delta_{\hat{p}}$, we can recover the corresponding $q \in \Delta_p$ via the inverse transformations. Thus, the mapping between $\Delta_p$ and $\Delta_{\hat{p}}$ is a bijection.

We can now prove the main result. By Theorem 4.1, we have

$$I_p(X_1, X_2; Y) = I_{\hat{p}}(\hat{X}_1, \hat{X}_2; \hat{Y}). \tag{81}$$

The PID solution in the original coordinates is given by minimizing the left-hand side of Equation (81) over $\Delta_p$, while the PID solution in the transformed coordinates minimizes the right-hand side over $\Delta_{\hat{p}}$. Because of the bijection between $\Delta_p$ and $\Delta_{\hat{p}}$, the optimization problems are equivalent, and the minimum values are the same. Therefore, the synergy values in both sets of coordinates are the same. We similarly conclude that all of the PID values are equal, as desired.

## B.3 Proof of Corollary 4.3

Since $q(x, y)$ is the PDF defined on the same space as $p(x, y)$, we have

$$I_p(X; Y) = \mathbb{E} \log \left[ \frac{p(x, y)}{p(x)p(y)} \right] = \mathbb{E} \log \left[ \frac{q(x, y)}{q(x)q(y)} \cdot \frac{p(x, y)}{q(x, y)} \cdot \frac{q(x)q(y)}{p(x)p(y)} \right] \tag{82}$$

$$= I_q(X; Y) + \mathbb{E} \log \left[ \frac{p(x, y)}{q(x, y)} \right] + \mathbb{E} \log \left[ \frac{q(x)}{p(x)} \right] + \mathbb{E} \log \left[ \frac{q(y)}{p(y)} \right] \tag{83}$$

$$= I_q(X; Y) + \mathrm{KL}\left( p(x, y) \| q(x, y) \right) - \mathrm{KL}\left( p(x) \otimes p(y) \| q(x) \otimes q(y) \right) \tag{84}$$

Firstly, since $\mathrm{KL}\left( p(x) \otimes p(y) \| q(x) \otimes q(y) \right) \geq 0$, we have

$$I_p(X; Y) - I_q(X; Y) \leq \mathrm{KL}\left( p(x, y) \| q(x, y) \right). \tag{85}$$

Secondly, given the monotonicity of the KL divergence, we have $\text{KL}\left(p(x,y)\|q(x,y)\right) \geq \text{KL}\left(p(x)\|q(x)\right)$ and $\text{KL}\left(p(x,y)\|q(x,y)\right) \geq \text{KL}\left(p(y)\|q(y)\right)$. Therefore, we have

$$I_p(X;Y) - I_q(X;Y) \geq \text{KL}\left(p(x,y)\|q(x,y)\right) - 2 \cdot \text{KL}\left(p(x,y)\|q(x,y)\right) \tag{86}$$
$$= -\text{KL}\left(p(x,y)\|q(x,y)\right) \tag{87}$$

Combining the two directions, we have

$$\left|I_p(X;Y) - I_q(X;Y)\right| \leq \text{KL}\left(p(x,y)\|q(x,y)\right). \tag{88}$$

## B.4 Proof of Proposition 4.4

Let $(\hat{X}, \hat{Y}) = f(X, Y) = f_X(X) \times f_Y(Y)$ be a Cartesian normalizing flow. Using the change of variables formula, we have

$$\log p(x,y) = \log p(\hat{x}, \hat{y}) + \log\left|\det \frac{\partial f(x,y)}{\partial(x,y)}\right|. \tag{89}$$

For the Cartesian flow $f = f_X \times f_Y$, the Jacobian is block-diagonal. Therefore, we have

$$\log\left|\det \frac{\partial f(x,y)}{\partial(x,y)}\right| = \log\left|\det \frac{\partial f_X(x)}{\partial x}\right| + \log\left|\det \frac{\partial f_Y(y)}{\partial y}\right|. \tag{90}$$

Given a dataset $\mathcal{D} = \{(x_1^{(j)}, x_2^{(j)}, y^{(j)}), j = 1, 2, \ldots, N\}$, note that maximizing the likelihood of $f_1(X_1) \times f_Y(Y)$ and $f_2(X_2) \times f_Y(Y)$ also minimizes $\text{KL}\left(p_{X_1 Y} \circ (f_1^{-1} \times f_Y^{-1})\|\mathcal{N}(\mu_1, \Sigma_{X_1 Y})\right)$ and $\text{KL}\left(p_{X_2 Y} \circ (f_2^{-1} \times f_Y^{-1})\|\mathcal{N}(\mu_2, \Sigma_{X_2 Y})\right)$. Therefore, the Gaussian marginal regularizer is equivalent to maximizing the log-likelihood of $\{(x_i^{(j)}, y^{(j)})\}_{j=1}^N$ such that $x_i^{(j)} = f_i^{-1}(\hat{x}_i^{(j)})$, $y^{(j)} = g^{-1}(\hat{y}^{(j)})$, where $(\hat{x}_i^{(j)}, \hat{y}^{(j)})$ are sampled from variational Gaussian distribution $\mathcal{N}(\mu_i, \Sigma_{X_i Y})$. Therefore, we use the maximum-likelihood estimates of $p(x_1, y)$ and $p(x_2, y)$ to regularize the supremum in Gaussian marginals

$$\mathcal{L}_\mathcal{N}(x_i^{(j)}, y^{(j)}) = \mathcal{L}_{\mathcal{N}(\mu_i, \Sigma_{X_i Y})}(\hat{x}_i^{(j)}, \hat{y}^{(j)}) + \log\left|\det \frac{\partial f_1(x_i)}{\partial x_i}\right| + \log\left|\det \frac{\partial f_Y(y)}{\partial y}\right|, \tag{91}$$

where $\mathcal{L}_{\mathcal{N}(\mu_i, \Sigma_{X_i Y})}(\hat{x}_i^{(j)}, \hat{y}^{(j)})$ is the likelihood of the latent multivariate Gaussian variables

$$\mu_i = (\mu_{\hat{X}_i}, \mu_{\hat{Y}}) = \left(\frac{1}{N}\sum_{j=1}^N f_i(x_i^{(j)}), \frac{1}{N}\sum_{j=1}^N f_Y(y_i^{(j)})\right), \tag{92}$$

$$\Sigma_{X_i Y} = \frac{1}{N}\sum_{j=1}^N \begin{pmatrix} f_i(x_i^{(j)}) - \mu_{\hat{X}_i} \\ f_Y(y^{(j)}) - \mu_{\hat{Y}} \end{pmatrix}\begin{pmatrix} f_i(x_i^{(j)}) - \mu_{\hat{X}_i} \\ f_Y(y^{(j)}) - \mu_{\hat{Y}} \end{pmatrix}^\top, \tag{93}$$

$$\mathcal{L}_{\mathcal{N}(\mu_i, \Sigma_{X_i Y})}(\hat{x}_i^{(j)}, \hat{y}^{(j)}) := -\frac{1}{2}\log|\Sigma_{X_i Y}| - \frac{1}{2}\left(x_i^{(j)} - \mu_i\right)^\top \Sigma_{X_i Y}^{-1}\left(x_i^{(j)} - \mu_i\right). \tag{94}$$

To simultaneously ensure that both $p(\hat{x}_1, \hat{y})$ and $p(\hat{x}_2, \hat{y})$ are approximately Gaussian, we train the flow using the loss function

$$\mathcal{L}_\text{flow}(\{X_1, X_2, Y\}) = \mathcal{L}_\mathcal{N}(\{(X_1, Y)\}) + \mathcal{L}_\mathcal{N}(\{(X_2, Y)\}), \tag{95}$$

where $\mathcal{L}_\mathcal{N}(\{(X_i, Y)\})$ is estimated by Monte Carlo sampling

$$\mathcal{L}_\mathcal{N}(\{(X_i, Y)\}) \approx \frac{1}{N}\sum_{j=1}^N \mathcal{L}_\mathcal{N}(x_i^{(j)}, y^{(j)}). \tag{96}$$

# C   Experimental details for synthetic datasets

We provide the experimental details of Section 5: data generation, feature processing, model architecture, and computing resources.

## C.1 Canonical Gaussian examples

**1D broadcast channel:** We illustrated results on canonical 1D Gaussian examples in Figure 1 and corroborate the correctness of Thin-PID by designing three cases:

*Unique and redundant information (left):*

$$Y \sim \mathcal{N}(0,1) \tag{97}$$
$$X_1 = Y + n_1, \qquad n_1 \sim \mathcal{N}(0,1), \qquad n_1 \perp\!\!\!\perp Y, \tag{98}$$
$$X_2 = X_1 + n_2, \qquad n_2 \sim \mathcal{N}(0,\sigma^2), \qquad n_2 \perp\!\!\!\perp X_1. \tag{99}$$

*Unique and synergistic information (middle):*

$$Y \sim \mathcal{N}(0,1) \tag{100}$$
$$X_1 = Y + n_1, \qquad n_1, n_2 \sim \mathcal{N}(0,\sigma^2), \qquad (n_1, n_2) \perp\!\!\!\perp Y, \tag{101}$$
$$X_2 = n_2, \qquad \mathrm{Corr}(n_1, n_2) = \rho. \tag{102}$$

*Redundant and synergistic information (right):*

$$Y \sim \mathcal{N}(0,1) \tag{103}$$
$$X_1 = Y + n_1, \qquad n_1, n_2 \sim \mathcal{N}(0,1), \qquad (n_1, n_2) \perp\!\!\!\perp Y, \tag{104}$$
$$X_2 = Y + n_2, \qquad \mathrm{Corr}(n_1, n_2) = \rho. \tag{105}$$

**Additional experiments on GPID at higher dimensionality:** We next design additional examples to validate Thin-PID at higher dimensionality, which are also benchmarks in [52].

**Case 1**: Let $X_{1,1} = \alpha Y_1 + n_{1,1}$, $X_{2,1} = Y_1 + n_{2,1}$, $X_{1,2} = Y_2 + n_{1,2}$, $X_{2,2} = 3Y_2 + n_{2,2}$, where $Y_1, Y_2, n_{1,i}, n_{2,i} \sim$ i.i.d. $\mathcal{N}(0,1)$, $i = 1, 2$. Here, $(X_{1,1}, X_{2,1}, Y_1)$ is independent of $(X_{1,2}, X_{2,2}, Y_2)$. Using the additive property, we are able to aggregate the PID values derived from their separate decompositions, each of which is associated with a known ground truth, as determined by the MMI-PID, considering $Y_1$ and $Y_2$ as scalars.

**Case 2**: Let $Y$ and $X_2$ be the same as in Case 1. Let $X_1 = H_1 R(\theta) Y$, where $H_1$ is a diagonal matrix with diagonal entries 3 and 1, and $R(\theta)$ is a $2 \times 2$ rotation matrix that rotates $Y$ at an angle $\theta$. When $\theta = 0$, $X_1$ has a higher gain for $Y_1$ and $X_2$ has higher gain for $Y_2$. When $\theta$ increases to $\pi/2$, $X_1$ and $X_2$ have equal gains for both $Y_1$ and $Y_2$ (barring a difference in sign). Since $(X_{1,1}, X_{2,1}, Y_1)$ is not independent of $(X_{1,2}, X_{2,2}, Y_2)$ for all $\theta$, we only know the ground truth at the endpoints.

**Results**. The results in Cases 1 and 2 are shown in Figure 3. The left and right subfigures show the PID values of different GPID algorithms, and the middle one shows the absolute error between each PID algorithm and the ground truth. We observe that Thin-PID achieves the best accuracy with the error $< 10^{-12}$, while the absolute error of Tilde-PID is $> 10^{-8}$.

**Case 3**: We test the stability of Thin-PID at a higher dimensionality of $d := d_{X_1} = d_{X_2} = d_Y$. We repeat the process of Case 1 and use additive property to concatenate $(X_1, X_2, Y)$. Therefore, the PID values should be doubled as we double the dimension $d$.

**Results**. The results of GPID with higher dimensionality are shown in Figure C.1. The PID values of Thin-PID match ground truth by doubling in value when the dimension of $(X_1, X_2, Y)$ doubles.

**Accuracy.** Figure C.2 shows the absolute errors in PID values using the Thin-PID algorithm. It is observed that the absolute error of the Thin-PID algorithm remains constrained below around $10^{-9}$, even as the dimensionality extends to 1024. However, the absolute error of Tilde-PID in [52] increases with increasing dimension and will exceed $10^{-5}$ when $d > 1024$. Therefore, Thin-PID is not only more efficient but also more accurate than Tilde-PID.

## C.2 Synthetic non-Gaussian examples

**Multivariate Gaussian with invertible nonlinear transformation:** We start with a multivariate Gaussian distribution $\mathcal{N}(0, \Sigma_{X_1 X_2 Y})$. The pointwise dataset $\{x_1^{(j)}, x_2^{(j)}, y^{(j)}\}_{j=0}^N$ is sampled from the joint Gaussian distribution. The "exact" truth of PID is obtained by calculating $H_1$, $H_2$, and $\Sigma_Y$ from $\Sigma_{X_1 X_2 Y}$ directly, then performing Thin-PID. To show the necessity of Flow-PID in non-Gaussian cases, we transform $x_1^{(j)}$, $x_2^{(j)}$, and $y^{(j)}$ into non-Gaussian distributions using three nonlinear

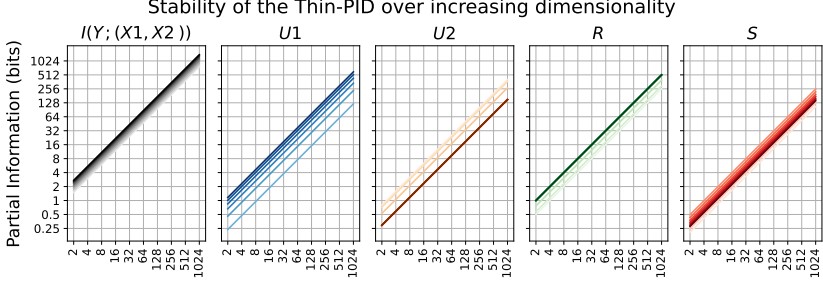

Figure C.1: GPID results when the dimension $d := d_{X_1} = d_{X_2} = d_Y$ increases. Different shadings represent different values of gain in $X_{1,1}(\alpha)$ in Case 1. The PID values of Thin-PID doubles every time when $d$ doubles.

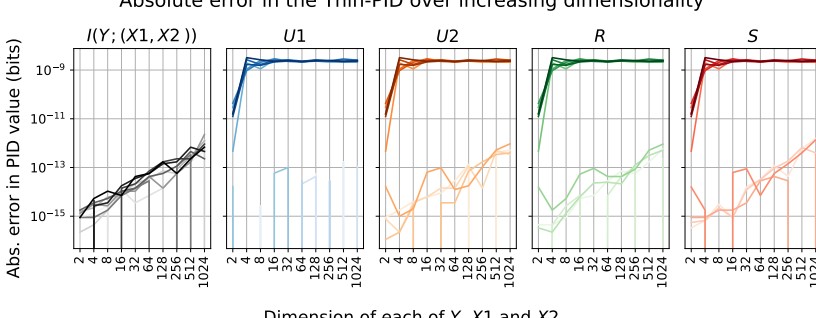

Figure C.2: Absolute errors of Thin-PID from Case 3.

invertible transformations $\widetilde{x_1} = (x_1)^3$, $\widetilde{x_2} = \sqrt[3]{x_2}$, and $\widetilde{y} = \sqrt[3]{y}$. According to Theorem 4.1, the MI of $(\widetilde{X_1}, \widetilde{X_2}, \widetilde{Y})$ remains the same as that of $(X_1, X_2, Y)$, but they are no longer pairwise Gaussian after the transformation.

Flow-PID learns the flow-based latent Gaussian encoder first, and then performs Thin-PID in the learned Gaussian marginal distributions. For Tilde-PID, the PID values are computed by directly estimating the covariance of non-Gaussian samples $(\widetilde{x_1}, \widetilde{x_2}, \widetilde{y})$. We did not include BATCH as a baseline in this case because BATCH requires feature clustering that is not feasible with a continuous target $y$. Therefore, BATCH cannot be generalized to regression or multitask scenarios where the target value is continuous.

The results are shown in Table 2. Flow-PID aligns with the exact truth in the relative PID values, whereas Tilde-PID fails with a distorted nature and degree of interactions.

Table C.1: Full results on non-Gaussian multi-dimensional examples.

| Dim | | $(2,2,2)$ | | | | $(10,5,2)$ | | | | $(30,10,2)$ | | |
|---|---|---|---|---|---|---|---|---|---|---|---|---|
| PID | $R$ | $U_1$ | $U_2$ | $S$ | $R$ | $U_1$ | $U_2$ | $S$ | $R$ | $U_1$ | $U_2$ | $S$ |
| Tilde-PID | 0.18 | 0.29 | 0.76 | 0.02 | 0.84 | 0 | 1.19 | 0.16 | 1.09 | 0 | 0.88 | 0.19 |
| Flow-PID | **0.62** | **0.91** | **0.50** | **0.11** | **2.36** | **0.32** | **0.19** | **0.45** | **2.18** | **1.13** | **0** | **0.17** |
| Truth | 0.79 | 1.46 | 0.58 | 0.18 | 2.96 | 0.54 | 0.26 | 0.58 | 2.92 | 2.18 | 0 | 0.25 |

| Dim | | $(100,60,2)$ | | | | $(256,256,2)$ | | | | $(512,512,2)$ | | |
|---|---|---|---|---|---|---|---|---|---|---|---|---|
| PID | $R$ | $U_1$ | $U_2$ | $S$ | $R$ | $U_1$ | $U_2$ | $S$ | $R$ | $U_1$ | $U_2$ | $S$ |
| Tilde-PID | 1.48 | 0 | 1.97 | 0.13 | 1.94 | 0 | 2.31 | 0.10 | 1.09 | 0 | 0.88 | 0.19 |
| Flow-PID | **4.34** | **0.36** | **0** | **0.25** | **3.79** | **0.15** | **0** | **0.36** | **0.48** | **0.98** | **0.47** | **0.11** |
| Truth | 5.71 | 1.01 | 0 | 0.57 | 7.85 | 0.14 | 0.05 | 0.90 | 0.81 | 1.45 | 0.58 | 0.19 |

**Specialized interactions with discrete targets:** We followed the settings of the synthetic generative model in [27]. Let $z_1, z_2, z_c \in \mathbb{R}^{50}$ be a fixed set of latent variables from $\mathcal{N}(0, \sigma^2)$ with $\sigma = 0.5$. $z_1, z_2, z_c$ represent latent concepts for the unique information of $X_1$, the unique information of $X_2$ and the common information, respectively. The concatenated variables $[z_1, z_c]$ are transformed into high-dimensional $x_1 \in \mathbb{R}^{100}$ using a fixed weight matrix $T_1 \in \mathbb{R}^{100 \times 100}$ and also $[z_2, z_c]$ to $x_2$ through $T_2$. The discrete label $y$ is generated by a function of $(z_1, z_2, z_c)$. By assigning different weights, the label $y$ can depend on (1) only $z_c$, which reflects pure redundancy, (2) only $z_1$ or $z_2$, which reflects pure uniqueness in $x_1$ or $x_2$, (3) the concatenation of $[z_1, z_2]$, which reflects pure synergy. More specifically,

$$X_1 = T_1 \cdot [z_1, z_c], \tag{106}$$

$$X_2 = T_2 \cdot [z_2, z_c], \tag{107}$$

$$Y = \left[ \text{sigmoid} \left( \frac{\sum_{i=0}^{n} f([z_1, z_2, z_c])_i}{n} \right) \geq 0.5 \right], \tag{108}$$

where $f$ is a fixed nonlinear transformation with dropout rate $p = 0.1$.

As shown in Table 3, we generate 10 synthetic datasets, including four specialized datasets $\{\mathcal{D}_R, \mathcal{D}_{U_1}, \mathcal{D}_{U_2}, \mathcal{D}_S\}$ with pure redundancy, uniqueness, or synergy. The rest are mixed datasets with $y$ generated from $(z_1, z_2, z_c)$ of different weights.

The ground-truth interactions are estimated by the test performance of multimodal models. The test accuracy $P_{\text{acc}}$ is converted to the MI between the inputs and the label using the bound:

$$I(X_1, X_2; Y) \leq \log P_{\text{acc}} + H(Y). \tag{109}$$

The information in each interaction is computed by dividing the total MI by the interactions involved in the data generation process: if the total MI is $0.6$ bits and the label depends on half of the common information between modalities and half from the unique information in $x_1$, then the ground truth $R = 0.3$ and $U_1 = 0.3$.

### C.3 Compute configuration and code Availability.

All experiments with synthetic datasets are performed on a Linux machine, equipped with 48GB RAM and NVIDIA GeForce RTX 4080.

The code used to reproduce results on synthetic datasets is included in a ZIP file as part of the supplementary material.

BATCH and Flow-PID require the training of neural networks (NNs). Before training NNs, we preprocess the feature by standardizing the features and randomly shuffling the mini-batches. BATCH follows the same training recipe as in [27], and the NN architectures used in Flow-PID are given in Table C.2.

Table C.2: The NN architectures for Flow-PID.

| NN | | Architecture |
|---|---|---|
| GLOW | $\times 1:$ | 4 (5) splits, 2 GLOW blocks between splits, |
| | | 16 hidden channels in each block, leaky constant $= 0.01$ |
| | $\times 1:$ | Orthogonal linear layer |
| | $\times 3:$ | RealNVP(AffineCouplingBlock(MLP(d/2, 64, d)), Permute-swap) |
| RealNVP | $\times 6:$ | RealNVP(AffineCouplingBlock(MLP($d/2$, 64, $d$)), Permute-swap) |

## D Experimental details for real-world datasets

We provide the experimental details of Section 6 and introduce the real-world datasets from Multi-Bench [30]. For the BATCH baseline, we follow the same experimental settings and training recipes in [27]. We release the data and code in an anonymous ZIP file attached with the supplementary materials.

Table C.3: Training recipe.

| Parameter | Value |
|---|---|
| Optimizer | Adam |
| Initial learning rate | 1e-4 |
| Scheduler | CosineAnnealingLR |
| Weight decay | 1e-4 |
| Data augmentation | Normalization |
| Batch size | 128 |

## D.1 Quantifying real-world datasets

**MultiBench datasets**: We use a collection of real-world multimodal datasets in MultiBench [30], which spans 10 diverse modalities (images, video, audio, text, time-series), 15 prediction tasks (humor, sentiment, emotions, mortality rate, ICD-9 codes, image-captions, human activities, digits, robot pose, object pose, robot contact, and design interfaces), and 5 research areas (affective computing, healthcare, multimedia, robotics, and HCI). These datasets are designed to test a combination of feature learning and arbitrarily complex interactions under different *multimodal fusion* models in the real world.

Table D.1: MultiBench datasets used for quantifying interactions between diverse modalities, tasks, and research areas.

| Datasets | Modalities | Size | Prediction task | Research areas |
|---|---|---|---|---|
| AV-MNIST [54] | {image, audio} | $60,000$ | logits | Multimedia |
| MOSEI [60] | {text, video, audio} | $22,777$ | sentiment, emotions | Affective Computing |
| UR-FUNNY [19] | {text, video, audio} | $16,514$ | humor | Affective Computing |
| MUSTARD [8] | {text, video, audio} | $690$ | sarcasm | Affective Computing |

**Real-world datasets with task-driven and causally relevant interactions**: we conducted 2 additional experiments with real-world data to demonstrate our method in additional application areas and with unexplored modalities.

- **TCGA-BRCA** is a multimodal dataset created to help study the causes and progression of breast cancer. We quantified the PID of predicting the breast cancer stage from protein expression and microRNA expression. Flow-PID **identified strong uniqueness for the modality of microRNA** expression as well as moderate amounts of redundancy and synergy. These results are also **in line with modern research**, which suggests microRNA changes as a direct result of cancer progression.
- **VQA** (Visual Question Answering) is a multimodal dataset consisting of 10,000 images with corresponding yes/no questions and their answers. Under the paradigm of using the image and question to predict the answer, one would naturally **expect high synergy since the image and question complement each other**. Our method, **Flow-PID, recovers exactly this** – we find synergy is the dominant interaction between the modalities.

The detailed experimental results are shown in Table D.2. Both of these experiments demonstrate further applications and additional modalities to validate our method.

Table D.2: Additional experimental results of Flow-PID on real datasets with task-driven interactions.

| Dataset | dim-$X_1$ | dim-$X_2$ | $R$ | $U_1$ | $U_2$ | $S$ | Expected interactions |
|---|---|---|---|---|---|---|---|
| VQA2.0 | 768 | 1000 | 0.22 | 0.26 | 0.0 | **0.76** | $S$ |
| TCGA | 487 | 1881 | 0.41 | 0.0 | **1.07** | 0.34 | $U_2$ |

**Information-preserving feature extractor**: For datasets with available modality features (images, text), we use the end-to-end PID estimator (Flow-PID, BATCH). For other modalities (audio, time-series), we first use pretrained encoders to extract features. To preserve the MI of the extracted

features, we add a contrastive loss to the encoder $\text{Enc}(X)$ using MINE [4]:

$$\mathcal{V}(\theta) = \frac{1}{N} \sum_{j=1}^{N} T_\theta \left( \text{Enc}(x^{(j)}), y^{(j)} \right) - \log \left( \frac{1}{N} \sum_{j=1}^{N} e^{T_\theta \left( \text{Enc}(x^{(k)}), \bar{y}^{(j)} \right)} \right). \tag{110}$$

### D.2  Model selection

**Setup**: Given a new dataset $\mathcal{D}$, we are interested in whether PID estimators are beneficial in recommending the most appropriate model *without training all models from scratch*. We hypothesize that the model with the best performance will likely perform well on the dataset most analogous to it, given the similarity in the interactions. Therefore, we select the most similar pre-trained dataset $\mathcal{D}^*$ from a set of base data sets $\mathcal{D}'$ (the 10 synthetic data sets presented in Table 3) by measuring the difference in the normalized PID values:

$$\mathcal{D}^* = \arg \min_{\mathcal{D}'} s(\mathcal{D}, \mathcal{D}') = \arg \min_{\mathcal{D}'} \sum_{I \in \{R, U_1, U_2, S\}} |I_\mathcal{D} - I_{\mathcal{D}'}|. \tag{111}$$

The quality of the model selection is evaluated by the percentage of the performance of the selected model with respect to the performance of the truly best-performing model on $\mathcal{D}$:

$$\% \text{ Performance}(f, f^*) = Acc(f)/Acc(f^*). \tag{112}$$

**Choices of multimodal models**: We implement 10 multimodal fusion models in 5 synthetic datasets and 5 MultiBench datasets.

1. ADDITIVE: Suitable unimodal models are first applied to each modality before aggregating the outputs using an additive average: $y = 1/2(f_1(\mathbf{x}_1) + f_2(\mathbf{x}_2))$ [20].
2. AGREE: Add another regularizer as prediction agreement $(+\lambda(f_1(\mathbf{x}_1) - f_2(\mathbf{x}_2))^2$ [10]).
3. ALIGN: Add feature alignment $(+\lambda \text{sim}(\mathbf{x}_1, \mathbf{x}_2)$ like contrastive learning [41]).
4. ELEM: Element-wise interactions for static interactions (i.e., without trainable interaction parameters): $y = f(\mathbf{x}_1 \odot \mathbf{x}_2)$ [1, 23].
5. TENSOR: Outer-product interactions (i.e., higher-order tensors): $y = f\left(\mathbf{x}_1 \mathbf{x}_2^\top\right)$ [14, 59, 21, 29, 33].
6. MI: Dynamic interactions with learnable weights include multiplicative interactions $\mathbf{W}$: $y = f(\mathbf{x}_1 \mathbf{W} \mathbf{x}_2)$ [24].
7. MULT: Dynamic interactions with learnable weights through cross-modal self-attention, which is used in multimodal transformers: $y = f(\text{softmax}(\mathbf{x}_1 \mathbf{x}_2^\top) \mathbf{x}_1)$ [51, 49, 57].
8. LOWER: Lower-order terms in higher-order interactions to capture unique information [33, 59].
9. REC: Reconstruction objectives to encourage maximization of unique information (i.e., adding an objective $\mathcal{L}_{\text{rec}} = \|g_1(\mathbf{z}_{\text{mm}}) - \mathbf{x}_1\|_2 + \|g_2(\mathbf{z}_{\text{mm}}) - \mathbf{x}_2\|_2$ where $g_1, g_2$ are auxiliary decoders mapping $\mathbf{z}_{\text{mm}}$ to each raw input modality [45, 50, 56].
10. EF (early fusion): Concatenating data at the earliest input level, essentially treating it as a single modality, and defining a suitable prediction model $y = f([\mathbf{x}_1, \mathbf{x}_2])$ [31].

**Architectures of multimodal models in different datasets**: To make a fair comparison between BATCH and Flow-PID, we adopt the same architecture of feature encoder, modality fusion, and training hyperparameters in [27]. For datasets with available modality features, we use data with standard pre-processing as the input of multimodal models. For other datasets without available modality features (UR-FUNNY, MUStARD, MOSEI), we first use pretrained encoders to extract features, which are also provided along with those datasets. The NN architectures and training hyperparameters are provided below.

Table D.3: NN architectures for multi-modal fusion models. The input dimension is decided by extracting $d$-dimensional features from the data. For datasets with available modality features, feature dim $d$ is identical to the data dim. For other datasets without available modality features, we first use pretrained encoders to obtain features with output dim $d$.

| Component | Model | Parameter | Value |
|---|---|---|---|
| Encoder | Identity | / | / |
| | Linear | Feature dim
Hidden dim | $[d, d]$
512 |
| Decoder | Linear | Feature dim
Hidden dim | $[d, d]$
512 |
| Fusion | Concat | / | / |
| | Elem
MI [24] | Output dim | 512 |
| | LOWER [33] | Output dim
rank | 512
32 |
| | MULT [49] | Embed dim
Num heads | 512
8 |
| Classification head | Identity | / | / |
| | 2-Layer MLP | Hidden size
Activation
Dropout | 512
LeakyReLU(0.2)
0.1 |
| Training | EF & ADDITIVE & ELEM & TENSOR
MI & MULT & LOWER | Loss
Batch size
Num epochs
Optimizer/Learning rate | Cross Entropy
128
100
Adam/0.0001 |
| | AGREE & ALIGN | Loss

Batch size
Num epochs
Optimizer/Learning rate
Cross Entropy Weight
Agree/Align Weight | Cross Entropy
+ Agree/Align Weight
128
100
Adam/0.0001
2.0
1.0 |
| | REC [50] | Loss

Batch size
Num epochs
Optimizer
Learning rate
Recon Loss Modality Weight
Cross Entropy Weight
Intermediate Modules | Cross Entropy
+ Reconstruction (MSE)
128
100
Adam
0.0001
$[1, 1]$
2.0
MLP $[512, 256, 256]$
MLP $[512, 256, 256]$ |

Table D.4: Table of hyperparameters for affective computing datasets.

| Component | Model | Parameter | Value |
|---|---|---|---|
| Encoder | Identity | / | / |
| | GRU | Input size
Hidden dim | $[5, 20, 35, 74, 300, 704]$
$[32, 64, 128, 512, 1024]$ |
| Decoder | GRU | Input size
Hidden dim | $[5, 20, 35, 74, 300, 704]$
$[32, 64, 128, 512, 1024]$ |
| Fusion | Concat | / | / |
| | Elem
MI [24] | Output dim | $[400, 512]$ |
| | Tensor Fusion [59] | Output dim | 512 |
| | MULT [49] | Embed dim
Num heads | 40
8 |
| Classification head | Identity | / | / |
| | 2-Layer MLP | Hidden size
Activation
Dropout | 512
LeakyReLU(0.2)
0.1 |
| Training | EF & ADDITIVE & ELEM & TENSOR
MI & MULT & LOWER | Loss
Batch size
Num epochs
Optimizer/Learning rate | L1 Loss
32
40
Adam/0.0001 |
| | AGREE & ALIGN | Loss

Batch size
Num epochs
Optimizer/Learning rate
Agree/Align Weight | L1 Loss
+ Agree/Align Weight
32
30
Adam/0.0001
0.1 |
| | REC [50] | Loss

Batch size
Num epochs
Optimizer
Learning rate
Recon Loss Modality Weight
Intermediate Modules | L1 Loss
+ Reconstruction (MSE)
128
50
Adam
0.001
$[1, 1]$
MLP $[600, 300, 300]$
MLP $[600, 300, 300]$ |

Table D.5: Table of hyperparameters for AV-MNIST encoders.

| Component | Model | Parameter | Value |
|---|---|---|---|
| Image Encoder | LeNet-3 | Filter Sizes
Num Filters
Filter Strides / Filter Paddings
Max Pooling | $[5, 3, 3, 3]$
$[6, 12, 24, 48]$
$[1, 1, 1, 1]$ /$[2, 1, 1, 1]$
$[2, 2, 2, 2]$ |
| Image Decoder | DeLeNet-3 | Filter Sizes
Num Filters
Filter Strides / Filter Paddings | $[4, 4, 4, 8]$
$[24, 12, 6, 3]$
$[2, 2, 2, 4]$/$[1, 1, 1, 1]$ |
| Audio Encoder | LeNet-5 | Filter Sizes
Num Filters
Filter Strides / Filter Paddings
Max Pooling | $[5, 3, 3, 3, 3, 3]$
$[6, 12, 24, 48, 96, 192]$
$[1, 1, 1, 1, 1, 1]$/$[2, 1, 1, 1, 1, 1]$
$[2, 2, 2, 2, 2, 2]$ |
| Audio Decoder | DeLeNet-5 | Filter Sizes
Num Filters
Filter Strides / Filter Paddings | $[4, 4, 4, 4, 4, 8]$
$[96, 48, 24, 12, 6, 3]$
$[2, 2, 2, 2, 2, 4]$/$[1, 1, 1, 1, 1, 1]$ |

Table D.6: Table of hyperparameters for ENRICO dataset in the HCI domain.

| Model | Parameter | Value |
|---|---|---|
| Unimodal | Hidden dim | 16 |
| MI-Matrix [24] | Hidden dim | 32 |
| | Input dims | 16, 16 |
| MI | Hidden dim | 32 |
| | Input dims | 16, 16 |
| Lower [33] | Hidden dim | 32 |
| | Input dims | 16, 16 |
| | Rank | 20 |
| Training | Loss | Class-weighted Cross Entropy |
| | Batch size | 32 |
| | Activation | ReLU |
| | Dropout | 0.2 |
| | Optimizer | Adam |
| | Learning Rate | $10^{-5}$ |
| | Num epochs | 30 |

