# OpenReview forum: "Partial Information Decomposition via Normalizing Flows in Latent Gaussian Distributions"
_NeurIPS.cc/2025/Conference — NeurIPS 2025 poster_

### Official Review · Reviewer_H15Z · 2025-06-30

**Clarity:** 3
**Significance:** 3
**Originality:** 3
**Rating:** 5
**Confidence:** 5

**Summary:**

## Summary
This paper introduces a new computational framework for Partial Information Decomposition (PID), focusing on continuous and high-dimensional multimodal data where traditional PID estimation methods struggle. The proposed method is a new efficient gradient-based algorithm for PID under Gaussian assumptions, and a flow-based extension to handle non-Gaussian, high-dimensional data.

**Questions:**

See Weakness

**Ethical Concerns:**

["NO or VERY MINOR ethics concerns only"]

**Final Justification:**

The proposed Partial Information Decomposition seem to have potential in many other machine learning areas due to its mathematical foundation. In the rebuttal stage, the author also addressed my concerns very timely and promised to release code for public research. So I consider giving rating 5 to accept the paper and encourage related research in the following years.

**Limitations:**

See Weakness

**Quality:**

3

**Strengths And Weaknesses:**

## Strength
1) This paper proves that when pairwise distributions are Gaussian, the optimal PID solution also lies within the space of joint Gaussians.
2) This paper demonstrates that PID profiles (from Flow-PID) can help select the best-performing multimodal fusion model for a new task.
3) The paper is well-formulated and well-grounded.
4) The results presented are good.

## Weakness:
1) Flow-PID’s accuracy depends on the quality of the Gaussianization step using normalizing flows.
2) Hard to rigorously validate PID estimates on real-world datasets where the true PID values are unknown.
3) Performance may degrade with highly complex feature distributions or small sample sizes.
4) Due to the fundamental aspect of the proposed method, it would be more impactful if the authors could open-source their code for people to draw connections between the maths and the implementation.

---

> ### Author Rebuttal · Authors · 2025-07-31
>
> We sincerely thank the reviewer for taking the time and effort to provide thoughtful reviews and insightful comments. We appreciate that the reviewer considers our work as a "**well-formulated and well-grounded paper**" with "**good presented results**". Below we will address your concerns point by point.
>
> > [R1] Flow-PID’s accuracy depends on the quality of the Gaussianization step using normalizing flows.
>
> We thank the reviewer for raising this concern. The pairwise Gaussianity is NOT a fundamental limitation. Modern normalizing flows (e.g., RealNVP, Glow) have been shown to approximate a wide class of continuous densities with sufficient depth and capacity. A large collection of experiments on synthetic datasets indicates that perfect Gaussianization is not required: we have empirically observed that moderate approximation suffices to recover meaningful and accurate PID decompositions.
>
> > [R2] Hard to rigorously validate PID estimates on real-world datasets where the true PID values are unknown.
>
> We thank the reviewer for raising this concern. Rigorous validation of PID estimates on real-world datasets is inherently difficult as the generation of multimodal data is unknown. However, we firstly want emphasize that **we have a large collection of synthetic datasets involving with diverse multimodalities**: Gaussian and non-Gaussian, discrete and continuous, low and high dimensional, and different specialized interactions, where the ground truth of PID is known. Our proposed model outperforms the SOTA baselines in all these synthetic cases, which partially shows its generalized capability on real datasets. Secondly, we used Flow-PID to select high-performance model by measuring the similarity between synthetic and real datasets, which helps support the correctness of PID estimates on complex real-world data. Lastly, we also provided 2 additional experiments on VQA2.0 and TCGA with task-driven interactions, which **provides evidence of interpretive alignment that are causally relevant**.
>
> | Dataset | dim-X1 | dim-X2 | R | U1 | U2 | S | Expected interactions |
> |------|-----|-----|-----|-----|-----|-----|-----|
> | TCGA | 487 | 1881 | 0.41 | 0.0 | **1.07** | 0.34 | U2 |
> | VQA2.0 | 768 | 1000 | 0.22 | 0.26 | 0.0 | **0.76** | S |
>
> > [R3] Performance may degrade with highly complex feature distributions or small sample sizes.
>
> We thank the reviewer for this raising this point. Both distributional complexity and limited data pose general challenges for information-theoretic estimators, including ours. Flow-PID specifically designed to address complex feature distributions by leveraging expressive normalizing flows. Regarding small sample sizes, we acknowledge that overfitting is a risk. To mitigate this, we employ a MI regularization in the encoder and flow. We listed this as one of limitations in Section 7, but we think this is a common challenge in PID society.
>
> > [R4] Due to the fundamental aspect of the proposed method, it would be more impactful if the authors could open-source their code for people to draw connections between the maths and the implementation.
>
> We thank the reviewer for this valuable feedback. We appreciate the concern regarding accessibility and agree that making the paper more approachable for a broader audience is important. Due to the anonymity, we only attached the anonymous code in the supplemental materials. **We will release the data and code for Thin-PID and Flow-PID to encourage further studies of multimodal information and modeling**.

---

> ### Comment · Reviewer_H15Z · 2025-08-05
>
> Most of my concerns have been addressed. Still, I would urge the open-source action, because there are always compromises between theoretical flow and the code. Some parts of the code may not strictly align with the proposed method or may have some flaws. But still, it leaves room for future researchers to think of and improve. I think this is very important when converting a mathematically focused concept to a running code in practice. However, I am not considering further raising score for the paper, because I think it is not a flawless paper yet.

---

> > ### Author Response · Authors · 2025-08-05
> >
> > We thank the reviewer for their engagement and appreciate this concern. **We have included anonymized code as part of our submission and commit to making it publicly available.** In the anonymized code, we have included the complete Thin-PID and Flow-PID code as well as scripts that we used to generate our results in Sections 5 and 6. In the open-source version, we will provide full documentation and tutorials to support reproducibility and future research.

---

### Official Review · Reviewer_18V1 · 2025-07-01

**Clarity:** 2
**Significance:** 2
**Originality:** 2
**Rating:** 4
**Confidence:** 2

**Summary:**

The paper proposes a novel partial information decomposition (PID) framework to quantify unique, redundant, and synergistic information in multimodal datasets, focusing on continuous and high-dimensional data. The main contributions include 1. Thin-PID, an efficient gradient-based Gaussian PID (GPID) algorithm that achieves linear complexity in some dimensions by reformulating the optimization problem and 2. Flow-PID, which uses normalized flow to transform non-Gaussian data into a latent Gaussian space for efficient PID estimation. The paper verifies the accuracy and efficiency of the method through experiments on synthetic datasets, and demonstrates its application value in tasks such as model selection on multiple real-world multimodal benchmarks.

**Questions:**

1.	In Flow-PID, when the KL divergence between the latent distribution and the Gaussian distribution is large (i.e., when the fit is poor), what kind of systematic deviation will the four components of PID (U1, U2, R, S) show? Is the collaborative information S more likely to be overestimated, or the redundant information R?

2.	In experiments on real-world datasets (Figure 4), the total mutual information estimated by Flow-PID is much higher than that of BATCH. In the absence of true labels, how can we be sure that this "higher" estimate is "more accurate" rather than "falsely high" due to overfitting the noise of training samples? Although the model selection experiment (Table 5) provides indirect evidence, is this conclusion robust enough?

3.	In Table 3, Flow-PID outperforms BATCH for discrete target tasks with collaborative information. Is this because Flow-PID’s Gaussian channel interpretation is naturally better at capturing this type of "XOR"-style interactions? Or is this perhaps an “inductive bias” introduced by the framework itself that favors the discovery of collaborative information?

**Ethical Concerns:**

["NO or VERY MINOR ethics concerns only"]

**Final Justification:**

The authors' rebuttal has addressed most of my concerns. I am keeping my original score.

**Limitations:**

yes

**Quality:**

2

**Strengths And Weaknesses:**

First of all, I would like to acknowledge that I am not familiar with the related works. My review and rating is conservative.

**Strength**

1.	The problem solved by the paper - designing an efficient PID estimator for high-dimensional continuous data - is of great significance in the fields of multimodal learning, explainable AI, etc. How to quantify and decompose information interaction is a cutting-edge and challenging topic.

2.	The paper cleverly combines PID theory, normalized flow, and the classic Gaussian channel model to form a novel Flow-PID framework. This fusion of cross-domain ideas provides a feasible engineering path to solve a difficult theoretical problem.

3.	Experiments on synthetic datasets and MultiBench show that Flow-PID outperforms baselines such as BATCH and CVX on high-dimensional and high-cooperative interaction data (Tables 2, 3, and 4).

**Weakness**

1.	The pairwise Gaussianity assumption (Definition 3.1) is too strict. Real multimodal data (such as text and images) often present complex non-Gaussian distributions, which may not be fully captured even by normalized flow transformation.

2.	The entire Flow-PID framework relies on a core assumption that any complex data distribution can be effectively mapped to a latent space with a Gaussian marginal distribution through normalized flow. The paper should further clarify the scope and limitations of this assumption.

3.	The technical details (such as normalized flow implementation and optimized reconstruction) are complex and difficult to understand for non-information theory PID experts, limiting the appeal of the paper to a wider audience.

---

> ### Author Rebuttal · Authors · 2025-07-31
>
> We sincerely thank the reviewer for taking the time and effort to provide thoughtful reviews and insightful comments. We appreciate that the reviewer consider our work as a "**novel PID framework**" and "**great significant in the fields of multimodal learning**" with theoretically and empirically demonstrated improvement. Below we will address your concerns point by point.
>
> > **[R1]** The pairwise Gaussianity assumption is too strict. Real multimodal data may not be fully captured even by normalized flow transformation.
>
> We appreciate the reviewer's insightful comment. The pairwise Gaussianity assumption is strong but NOT a huge limitation. To clarify, our use of Definition 3.1 is not a requirement for Flow-PID, but rather serves to motivate and analyze the limitations of existing GPID estimators like Tilde-PID. The very motivation for developing Flow-PID is to relax this assumption and enable decomposition under arbitrary non-Gaussian continuous multimodal distributions. Regarding the concern that normalizing flows may still be insufficient to capture highly complex real-world data:
>
> - Flexible approximation beyond Gaussianity: Modern normalizing flows (e.g., RealNVP, Glow) have been shown to **approximate a wide class of continuous densities with sufficient depth and capacity**. In practice, we find that even moderately expressive flows (3-5 layers) already are capable of capturing skewed or heavy-tailed marginals.
> - Empirical validation: We have **a large selection of different synthetic data with known ground truth** designed with high-dimensional input modalities, non-Gaussian transformations, and different specialized interactions. **Flow-PID outperforms existing PID estimators in all these non-Gaussian cases** (*Table 2, 3, C.1*), which shows that normalizing flows can capture the appropriate latent marginal Gaussian distributions we need.
> - Future extensions: We thank the reviewer for raising the point that even flow-based models may have limitations in extremely high-dimensional or discrete-continuous hybrid settings. We view Flow-PID as a step forward toward tractable, expressive, and general-purpose PID estimation. Future work can explore more advanced generative models (e.g., latent diffusion models, transformers) to produce MI-preserved encoders in multimodal learning.
>
> > **[R2]** The entire Flow-PID framework relies on a core assumption that any complex data distribution can be effectively mapped to a latent space with a Gaussian marginal distribution through normalized flow.
>
> We thank the reviewer for this important observation. We acknowledge that Flow-PID assumes the existence of an invertible mapping from complex data distributions to a Gaussian latent space, and we agree that the scope and limitations of this assumption warrant further clarification, which is listed as one of the limitations in Section 7. However, **this assumption is NOT a significant limitation**, and it does NOT prevent us from estimating PID in latent Gaussian distributions.
>
> - Gaussianity assumption as a practical modeling tool, not a theoretical requirement: the use of normalizing flows in Flow-PID is not to enforce perfect Gaussianity, but rather to enable a tractable surrogate space where gradient-based optimization and Thin-PID become more manageable on continuous data and targets.
> - Expressive power of normalizing flows: Our results show that the approximation error is empirically tolerable since Flow-PID is capable of dealing with real-world features of dimension 512, 1000, 1881, etc.
> - By Optimal Transport (OT) theory, the existence of a flow that transport from a source distribution to the target distribution is guaranteed under fairly mild conditions.
>
> > **[R3]** The technical details (such as normalized flow implementation and optimized reconstruction) are complex and difficult to understand for non-information theory PID experts, limiting the appeal of the paper to a wider audience.
>
> Thank you for this valuable feedback. We appreciate the concern regarding accessibility and agree that making the paper more approachable for a broader audience, including those not deeply familiar with information-theoretic PID or normalizing flows, is important. Due to the page limitation, **we put the technical details in the Appendix A and B**. Additionally, **we uploaded the anonymous code and provided the settings and hyperparameters in Appendix C and D for reproducibility**. We will release the data and code for Thin-PID and Flow-PID to encourage further studies of multimodal information and modeling.
>
> > **[R4]** In Flow-PID, when the KL divergence between the latent distribution and the Gaussian distribution is large (i.e., when the fit is poor), what kind of systematic deviation will the four components of PID (U1, U2, R, S) show? Is the collaborative information S more likely to be overestimated, or the redundant information R?
>
> We thank the reviewer for this thoughtful and technically nuanced question. Empirically when the KL divergence is large, we observe the following tendencies:
>
> - Redundancy (R) tends to be overestimated when the flow fails to decorrelate shared structure between sources. This happens when high-order correlation gets incorrectly interpreted as mutual redundancy in a near-Gaussian latent space.
> - Synergy (S) is often underestimated in such cases, because synergy requires modeling conditional interactions that are especially sensitive to misrepresentation of joint dependencies.
> - Uniqueness (U) may show mild variability, but could be biased if the flow introduces asymmetries between sources (e.g., when one source is mapped more accurately than the other).
>
> In our experiments on real dataset, the approximation error is inevitable but empirically tolerable. Flow-PID are robust to moderate approximation error and can quantify specialized interactions as we expected.
>
> > [R5] In experiments on real-world datasets (Figure 4), the total mutual information estimated by Flow-PID is much higher than that of BATCH. In the absence of true labels, how can we be sure that this "higher" estimate is "more accurate" rather than "falsely high" due to overfitting the noise of training samples? Although the model selection experiment (Table 5) provides indirect evidence, is this conclusion robust enough?
>
> Thank you for raising this question. First of all, we want to emphasize that we have a large collection of synthetic datasets, involving with diverse multimodalities: Gaussian and non-Gaussian, discrete and continuous, low and high dimensional, and different specialized interactions, where the ground truth of PID is known. Our Flow-PID outperforms the SOTA baselines
> in all these synthetic cases, which partially shows its generalized capability on real datasets. As for the total mutual information, we can provide some more clarity on this claim and provide more quantitative evidence (although we are unable to provide visualization due to the change in format). When we are able to predict the output with very low probability of error, we expect the mutual information to be very close to the entropy of the output (in fact, for perfect prediction, the mutual information is precisely the entropy of the output). However, with high probability of error, we expect a lower mutual information. As is done in the work by Liang et al., we can also estimate the mutual information via $I(Y; X_1, X_2)$ by $H(Y) + \log P_{\text{acc}}$, which is  a bound on the mutual information due to Feder and Merhav [1]. This estimate matches much more closely with the total mutual information determined by Flow-PID rather than BATCH. We see that BATCH tends to recognize much less mutual information, possibly due to it being highly dependent on the accuracy of unimodal and multimodal classifiers. We find this to be an indirect verification that our method is able to identify interactions better than previous works. Lastly, Flow-PID selects the most appropriate model by measuring the similarity between synthetic and real datasets. Given that fact that Flow-PID performs well on a large collection of synthetic datasets, good model selection results infer the appropriate PID estimates on complex real-world data.
>
> [1] Meir Feder and Neri Merhav. Relations between entropy and error probability. IEEE Transactions on Information theory, 40(1):259–266, 1994.
>
> > [R6] In Table 3, Flow-PID outperforms BATCH for discrete target tasks with collaborative information. Is this because Flow-PID’s Gaussian channel interpretation is naturally better at capturing this type of "XOR"-style interactions? Or is this perhaps an “inductive bias” introduced by the framework itself that favors the discovery of collaborative information?
>
> Thank you for this excellent question. Theoretically, the Gaussian channel interpretation allows us to solve PID exactly in marginal Gaussian distributions. Empirically, the flow-based encoder has expressive power of learning a wide class of continuous transformations, and Flow-PID are robust to moderate approximation error. On the other hand, BATCH approximates the pointwise discrete $p(x_1,x_2,y)$ using Sinkhorn's algorithm, which may introduce large approximation error. Additionally, it is known that BATCH generally underestimates synergy [2] since it is directly dependent on estimation of $p(y | x_1, x_2)$, which is done via the best performing multimodal classification model. Therefore, it is shown in Table 3 that Flow-PID performs better on tasks involving strong synergistic information, such as XOR-style dependencies. We also want to note that Flow-PID also outperforms BATCH in tasks with specialized redundancy and uniqueness.
>
> [2] Liang, P. P., Cheng, Y., Fan, X., Ling, C. K., Nie, S., Chen, R., Deng, Z., Allen, N., Auerbach, R., Mahmood, F., et al. (2024). Quantifying \& modeling multimodal interactions: An information decomposition framework. Advances in Neural Information Processing Systems, 36.

---

> > ### Comment · Reviewer_18V1 · 2025-08-05
> >
> > The authors' rebuttal has addressed most of my concerns. I am keeping my original score.

---

> > > ### Author Response · Authors · 2025-08-07
> > >
> > > We thank the reviewer for the constructive comments, which are valuable in helping us strengthen our work. We will incorporate your suggestions into the revised manuscript.

---

### Official Review · Reviewer_Qrhw · 2025-07-02

**Clarity:** 2
**Significance:** 3
**Originality:** 3
**Rating:** 4
**Confidence:** 4

**Summary:**

The paper proposes a new framework Flow-PID along with a new PID optimization objective (Thin-PID) to estimate the interactions between modalities in multimodal learning. These modules come along with theoretical analysis on the complexity of the proposed algorithm and the optimality of Gaussian joint distribution restriction. The authors provide empirical evaluation on both synthetic data and real-world multimodal data, comparing the proposal with other PID estimation approaches.

**Questions:**

I’m wondering about the originality and novelty of the paper. Despite soundness and strong foundation, the proposal seems to be a combination of existing solutions for related problems without significant improvements.

**Ethical Concerns:**

["NO or VERY MINOR ethics concerns only"]

**Final Justification:**

Based on the original submission and the authors’ rebuttal, I am inclined to recommend acceptance of this paper.

**Limitations:**

See weaknesses

**Quality:**

3

**Strengths And Weaknesses:**

### Strengths:
- The idea of transforming to latent distributions using normalizing flow, then optimizing to estimate PID is quite interesting.
- The authors provide formal definition of the problem and explanation of essential derivation steps.
- The proposal is built upon strong theoretical works, without loss of optimality guaranteed.
### Weaknesses:
- Major
   - The overall flow of the proposal, while being sound, is presented in a disjointed style. The reviewer assumes that the framework can be trained end-to-end, since the gradient of the Thin-PID can be estimated directly (Proposition 3.4), while Sec.4 and Fig.1 likely indicate the separated training (2 phases). If so, why is end-to-end training not possible ? If not, why is end-to-end training better than separated training ?
   - The Thin-PID, one of the main contributions, is directly built upon Tilde-PID[1], where the additional points are discussion on validity of Gaussian assumptions[1] and a different way of computing gradients. While both of them have strong foundations and some impacts, the reviewer does not see the significant contributions to the Tilde-PID. The optimization problem and solution approach are quite similar to the original work [1] also.
   - The second contribution, Flow-PID, combines MI-preserved encoder and ThinPID algorithm to extend the PID estimation to arbitrary distributions of $X_1$, $X_2$, Y. Here, the proposed MI-preserved encoder idea is quite similar to an existing work[2], which applies for a variable pair $(X; Y)$ only. This paper likely extends this framework to multimodal settings by processing $(X_1; X_2; Y)$ as $p(X_1, Y)$ and $p(X_2, Y)$, which is quite straightforward.
- Minor:
   - In Sec. 3.2, the author should provide definitions of $H_1, H_2$ for clarity.
   - There should be reference to Appendix in all derivation steps, the derivations in main text are quite ambiguous.
   - Can the proposal be used to estimate PID for a dataset with more than 2 modalities ? If so, the author should provide the results, e.g., estimating $U_1, U_2, U_3$ and so on.
   - For the claim in lines 295-296, can the author provide more results or visualization to show the correlation between performance and estimated PID ? This observation could be an interesting point.

[1] Praveen Venkatesh, Corbett Bennett, Sam Gale, et al. Gaussian Partial Information Decomposition: Bias Correction and Application to High-dimensional Data. Advances in Neural Information Processing Systems 2023.

[2] Butakov I. D., Tolmachev A. D., Malanchuk S. V., Neopryatnaya A. M., Frolov A. A., Mutual Information Estimation via Normalizing Flows, Advances in Neural Information Processing Systems 2024.

---

> ### Author Rebuttal · Authors · 2025-07-31
>
> We sincerely thank the reviewer for taking the time and effort to provide thoughtful reviews and insightful comments. We appreciate that you think our proposed Flow-PID and Thin-PID is "**quite interesting**" and "**new**" built upon "**strong theoretical works**". Below we will address your concerns point by point.
>
> > **[R1]** The overall flow of the proposal, while being sound, is presented in a disjointed style. End-to-end training or separated training? Which is better?
>
> We thank the reviewer for this insightful concern, and this is closely related to our contributions. Below we will address your concern in detail.
> - The flow-based MI-preserved encoder can NOT be trained end-to-end with Thin-PID, and **they should be optimized in two stages**. Thin-PID and flow-based encoder use different strategies. Note that Thin-PID is to **solve an optimization problem with NO need to train a NN model**. Although Thin-PID uses gradient descent to find the optimal solution, it does not involve with training another NN-based model to infer the optimal solution. The advantage is that Thin-PID only takes less than 20s to search the optimal solution even when the dimension of the modality is as large as 1000.
> - **It is meaningless to perform Thin-PID when the flow is not sufficiently trained**. Note that Thin-PID is exact when the pairwise marginals $p(x_i,y)$ are Gaussian. Hence, there's no need to perform Thin-PID when the flow is not sufficiently trained in a good manner. Although Thin-PID itself is very efficient, **we still only need to perform Thin-PID once** after we transform the input into a latent marginal Gaussian space with a sufficiently trained MI-preserved encoder.
> - This separate optimization strategy **allows us to reuse pretrained encoders** to learn latent Gaussian distributions, which will significantly enhance the efficiency of Flow-PID and make it much more faster than existing PID estimators (e.g., CVX, BATCH).
>
> We sincerely thank the reviewer for pointing out the ambiguity, and we will revise it in the final manuscript.
>
> > **[R2]** The Thin-PID, one of the main contributions, is directly built upon Tilde-PID [1], where the additional points are discussion on validity of Gaussian assumptions [1] and a different way of computing gradients. ... The optimization problem and solution approach are quite similar to the original work [1] also.
>
> We appreciate the reviewer’s thoughtful comparison to Tilde-PID [1], and we would like to clarify the key technical and conceptual contributions of Thin-PID beyond that prior work.
> - **Our first contribution is to rigorously prove that GPID can be solved by a joint Gaussian distribution without loss of optimality**. Tilde-PID assumes the optimal solution of GPID is a joint Gaussian distribution, which leaves this an open question. On the other words, **Thin-PID is exact**, while Tilde-PID only solves an optimizing bound on true PID values.
> - **RProp and bias correction in [1] are generalized methods** which can be applied in any gradient-based and sampling-based optimization problems. We acknowledge that our Thin-PID is inspired by the idea of GPID proposed in [1], and we borrowed the RProp and bias correction from Tilde-PID. During our experimental tests, we observed that RProp is indeed helpful for convergence. Therefore, we give them the credits here.
> - **The objective function and derivation of gradients in Thin-PID are completely different** from those in Tilde-PID, although both Thin-PID and Tilde-PID solve the GPID problem $\min I(X_1,X_2;Y)$. We use the same set of notations to provide a clear and unified understanding of the GPID problem. But **the design of the Thin-PID algorithm is new**. Please see Appendix A.2 for a detailed derivation, which is completely different from Tilde-PID.
> - Thanks to our newly designed objective function in Theorem 3.3, Thin-PID enjoys many advantages against Tilde-PID: 1) **Thin-PID has better accuracy**, which is shown in Figure C.1 and C.3 in Appendix C; 2) **Thin-PID has lower computational complexity**, which is shown in Table 1 and Figure 3; 3) **Thin-PID is more stable** when the dimensions of $X_1$ and $X_2$ are large.
>
> [1] Venkatesh, et al. Gaussian Partial Information Decomposition: Bias Correction and Application to High-dimensional Data, 2023.
>
> > **[R3]** The second contribution, Flow-PID, combines MI-preserved encoder and Thin-PID algorithm to extend the PID estimation to arbitrary distributions. Here, the proposed MI-preserved encoder idea is quite similar to an existing work.
>
> We thank the reviewer for raising this point. We appreciate the opportunity to clarify both the novelty and motivation behind our MI-preserved encoder within the Flow-PID framework. While the idea of preserving MI during representation learning has been explored in prior work, our application and integration in PID estimates is fundamentally different in several aspects:
> - **Targeted for PID-specific inference**: Our encoder is explicitly designed to preserve task-relevant information for PID, not for classification or MI estimation. We preserve MI between two modalities and the target, which is **crucial for estimating unique and redundant information components**. Prior flow-based encoders do not target this objective and are **not suitable out-of-the-box for PID estimation**.
> - Architectural coupling with Thin-PID and Flow modeling: In Flow-PID, the encoder is not an isolated module but a critical component that interfaces with the underlying optimization problem and surrogate gradient estimator in Thin-PID. We want to note that **our goal is NOT to propose a new MI estimator, but to extend PID estimation to arbitrary (non-Gaussian, continuous, multimodal) distributions** - a capability lacking in existing MI-based encoders.
> - New theorems on estimating PID with MI-preserved encoders: we develop a new underlying optimization problem in Flow-PID to minimize the KL divergence between variational Gaussian marginals $q(x_i,y)$ and the true marginals with invertible flow-based transformations. Although approximating MI with normalizing flows [2] has been explored, **tailoring a flow-based encoder to fit for PID estimates** is not trivial, and this is a novel framework.
>
> [2] Butakov, et al, Mutual Information Estimation via Normalizing Flows, 2024.
>
> > **[R4]** In Sec. 3.2, the author should provide definitions of $H_1$, $H_2$ for clarity.
>
> Thank you for this suggestion. We will add a clear pointer to Appendix. Since $(X_1, Y)$ is Gaussian in GPID, we can derive the matrix $H_1$ directly from the covariance matrix of $(X_1, Y)$. Similarly, $H_2$ is derived directly from the covariance matrix of $(X_2, Y)$. The detailed interpretation is provided in Appendix A (line 46).
>
> > **[R5]** There should be reference to Appendix in all derivation steps, the derivations in main text are quite ambiguous.
>
> Thank you for this suggestion. We include the proofs of theorems and propositions in the appendix. Due to the limited space of the manuscript, we did not include enough clear pointers in the main text. But we will fix this to make sure the audience can have clear access to detailed derivations.
>
> > **[R6]** Can the proposal be used to estimate PID for a dataset with more than 2 modalities?
>
> While the PID framework is defined for 2 modalities, it is common to evaluate the PID on each pair of modalities, and Flow-PID can easily be extended for this. We can jointly train an MI-preserved encoder for each modality according to the analogous Gaussian constraints, and then run Thin-PID on each pair of modalities. In fact, since Thin-PID is very efficient, our approach would allow us to compute the PID on several pairs of modalities much more efficiently than BATCH/CVX. We appreciate that the reviewer raises this interesting question, which can future work in this area.
>
> > **[R7]** For the claim in lines 295-296, can the author provide more results or visualization to show the correlation between performance and estimated PID? This observation could be an interesting point.
>
> Thank you for this question. We provide more clarity on this claim (although we are unable to provide visualization due to the change in format). When we are able to predict the output with very low probability of error, we expect the mutual information to be very close to the entropy of the output (in fact, for perfect prediction, the mutual information is precisely the entropy of the output). However, with high probability of error, we expect a lower mutual information. As is done in the work by Liang et al., we can also estimate the mutual information via $I(Y; X_1, X_2)$ by $H(Y) + \log P_{\text{acc}}$, which is  a bound on the mutual information due to Feder and Merhav [3]. This estimate matches much more closely with the total mutual information determined by Flow-PID rather than BATCH. We see that BATCH tends to recognize much less mutual information, possibly due to it being highly dependent on the accuracy of unimodal and multimodal classifiers. We find this to be an indirect verification that our method is able to identify interactions better than previous works.
>
> [3] Feder, et al. Relations between entropy and error probability, 1994.
>
> > Question: Originality and novelty . Despite soundness and strong foundation, the proposal seems to be a combination of existing solutions for related problems without significant improvements.
>
> Thank you for raising this concern. We appreciate the opportunity to clarify our novelty and motivation. **We have detailed responses in [R2] and [R3]**. Here we summarize our contributions and novelty:
> - Solve an open problem in GPID by proving the optimality of joint Gaussian solutions
> - The design of Thin-PID algorithm is novel, with a newly designed objective and less complexity
> - Flow-PID is a novel framework which extends GPID to arbitrary multimodal distributions
> - Compared to other PID algorithms: more accurate, more efficient, more generalized

---

> > ### Comment · Reviewer_Qrhw · 2025-08-04
> > **Official comment of Reviewer Qrhw**
> >
> > Dear Authors,
> >
> > Thanks for the detailed rebuttal.
> >
> > Most of my concerns have been cleared by now. Please consider reflecting these clarifications to the revised manuscript suitably.
> >
> > Since you have addressed my concerns, I raise my score from 3 to 4 to show my support for the paper.
> >
> > Regards.

---

> > > ### Author Response · Authors · 2025-08-05
> > >
> > > We appreciate the reviewer’s support and are glad that the concerns have been addressed. We will incorporate appropriate clarifications into the revised manuscript.

---

### Official Review · Reviewer_SRhn · 2025-07-08

**Clarity:** 2
**Significance:** 2
**Originality:** 3
**Rating:** 4
**Confidence:** 3

**Summary:**

This paper presents a novel framework for Partial Information Decomposition (PID) that efficiently quantifies information interactions in multimodal data. The authors introduce **Thin-PID**, a gradient-based algorithm that achieves 10× speedup for Gaussian PID computation, and **Flow-PID**, which uses normalizing flows to transform arbitrary distributions into latent Gaussian spaces while preserving information content. By proving the optimality of joint Gaussian solutions and developing information-preserving encoders, this work overcomes the computational bottleneck of existing PID methods for high-dimensional continuous data. Extensive experiments on synthetic and real-world multimodal benchmarks demonstrate superior accuracy and efficiency compared to state-of-the-art baselines.

**Questions:**

Given that baseline values ​​of PID in real datasets are unknown, how can we rigorously verify that Flow-PID produces meaningful and accurate decompositions, rather than just computationally convenient decompositions? The model selection task only provides indirect validation - can you propose more direct evaluation metrics or synthetic-to-real transfer experiments to better demonstrate the correctness of your PID estimates on complex real-world data?

**Ethical Concerns:**

["NO or VERY MINOR ethics concerns only"]

**Final Justification:**

My issue has been largely resolved, so I'm considering raising my rating.

**Limitations:**

Yes.

**Paper Formatting Concerns:**

No.

**Quality:**

3

**Strengths And Weaknesses:**

**Strengths**

1. The authors provide rigorous mathematical proofs, particularly establishing the optimality of joint Gaussian solutions for GPID, which resolves a previously open theoretical question and ensures their approach doesn't sacrifice accuracy for efficiency.
2. Thin-PID achieves over 10× speedup compared to state-of-the-art methods by cleverly reducing the optimization problem to a lower-dimensional space, making PID computation feasible for high-dimensional real-world applications.
3. Flow-PID ingeniously uses normalizing flows as "information-preserving translators" to handle non-Gaussian data, extending the benefits of efficient Gaussian computation to arbitrary distributions without information loss.
4. The paper demonstrates effectiveness across multiple scenarios - from synthetic data with known ground truth to diverse real-world multimodal benchmarks, showing both accuracy improvements and practical utility in model selection tasks.

**Weaknesses**

1. Although the authors demonstrate a 10× speedup on synthetic data, there is a lack of detailed computational time comparisons on real multimodal datasets, especially when the data dimensionality is really high.
2. The real data experiments are indeed insufficient - they are only tested on a few MultiBench datasets and lack verification in a wider range of real application scenarios.
3. I think the training complexity of Flow-PID is underestimated, the training of the normalized flow itself can be expensive and unstable, and the authors do not fully discuss this overhead, which may offset the speed advantage of Thin-PID.
4. The limitations of the Gaussian approximation assumption in the article are not discussed enough. Although the normalized flow is information-preserving in theory, there is a lack of in-depth analysis of how the approximation error in actual training affects the final PID estimation.
5. I look at the MultiBench datasets and find that many of them may not be truly "high-dimensional". For example, the feature dimensions after being processed by the pre-trained encoder may have been reduced, which weakens the persuasiveness of the "high-dimensional advantage".

---

> ### Author Rebuttal · Authors · 2025-07-31
>
> We sincerely thank the reviewer for taking the time and effort to provide thoughtful reviews and insightful comments. We appreciate that you consider our work as a "**novel framework**" and find our work "**mathematically rigorous**" with "**demonstrated effectiveness across multiple scenarios**". Below we will address your concerns point by point.
>
> > **[R1]** Although the authors demonstrate a 10× speedup on synthetic data, there is a lack of detailed computational time comparisons on real multimodal datasets.
>
> Thank you for this valuable feedback. Below we clarify the efficiency of our method compared with existing methods in detail.
> - **Reduced computational time results from the dimension, not the feature itself**: Thin-PID has less computational complexity of $O(\min(d_{X_1}, d_{X_2}))$, which is much less than $O((d_{X_1}+d_{X_2}))$ in Tilde-PID. *Figure 3* corroborates this improvement by showing that the empirical execution time of Thin-PID is around $10\times$ faster than Tilde-PID if they have the same input dimensions. Therefore, **Thin-PID still outperforms Tilde-PID in real datasets if they have the same feature dimension**. For example, when the input dimension is 1000, the execution time of Thin-PID is only 20s, while Tilde-PID takes more than 200s.
> - **Thin-PID is also more stable and accurate** than Tilde-PID. If the dimension of the input modality is larger than 1000, Tilde-PID struggles to converge and can be incomputable, while our proposed Thin-PID can handle dimensions more than 1000. As shown in *Figure C.1* and *C.2* (*Appendix C*), the accuracy of Thin-PID is better than Tilde-PID, especially when the dimension is high.
>
> > **[R2]** The real data experiments are indeed insufficient - they are only tested on a few MultiBench datasets and lack verification in a wider range of real application scenarios.
>
> Thank you for raising this concern. We selected the MultiBench benchmark due to its diversity in modality combinations and tasks, which we believe **cover a representative spectrum of multimodal learning scenarios and highlight the strength of our model**.
>
> **The modalities of the chosen datasets are diverse**, including text, vision, audio, physiology, et al. The dimension of the latent Gaussian features in UR-FUNNY, MUStARD, and MOSEI is 512, which is much higher than $53\pm 16$ PCA components used in Tilde-PID. To further support the applicability of Flow-PID in diverse scenarios, **we provide 2 additional experiments** on TCGA (Cancer genome atlas) and VQA2.0 (visual question answering) with causally relevant interactions.
>
> - **TCGA-BRCA**: we quantified the PID of predicting the breast cancer stage from protein expression and microRNA expression. Flow-PID **identified strong uniqueness for the modality of microRNA expression** as well as moderate amounts of redundancy and synergy. These results are also **in line with modern research** which suggests microRNA changes as a direct result of cancer progression.
> - **VQA**: consists of 10,000 images with corresponding yes/no questions and their answers. Under the paradigm of using the image and question to predict the answer, one would naturally **expect high synergy since the image and question complement each other**. Our method, **Flow-PID**, **recovers exactly this** -- we find synergy is the dominant interaction between the modalities.
>
> The detailed results are shown in the table below. Both experiments demonstrate further applications and additional modalities to validate our method.
>
> | Dataset | dim-X1 | dim-X2 | R | U1 | U2 | S | Expected interactions |
> |------|-----|-----|-----|-----|-----|-----|-----|
> | TCGA | 487 | 1881 | 0.41 | 0.0 | **1.07** | 0.34 | U2 |
> | VQA2.0 | 768 | 1000 | 0.22 | 0.26 | 0.0 | **0.76** | S |
>
> > **[R3]** I think the training complexity of Flow-PID is underestimated, the training of the normalized flow itself can be expensive and unstable, and the authors do not fully discuss this overhead, which may offset the speed advantage of Thin-PID.
>
> Thank you for raising this important point, which is closely related to our motivations. We agree that the training cost of normalizing flows could be significant in general. However, this will NOT offset the speed advantage of Thin-PID. On the contrary, **the very fast speed of Thin-PID (which is less than 20s with dimension 1000) motivates us to train a MI-preserved encoder** (typically, a normalizing flow) to transform the multi-modalities into latent Gaussian distributions so that PID can be solved efficiently in continuous and high-dimensional space.
>
> - **An MI-preserved encoder is necessary** in any case if you want to solve PID in high-dimensional space. The SOTA discrete PID estimator, BATCH, requires to learn unimodal encoders $p(y|x_i)$ first to estimate the marginal distribution of $p(x_i,y)$. On the other hand, GPID can be solved in a continuous space, but it restricts Gaussian marginals of $p(x_i,y)$. Hence, **there's no free lunch** if we want to solve PID in a continuous, high-dimensional, and non-Gaussian space.
>
> - **Flow-PID can reuse pretrained encoders**, while existing PID algorithms (e.g., BATCH) requires additional training to estimate $p(x_i)$. Compared to the post-training of BATCH, the optimization of Thin-PID is much more efficient.
>
> - Modern normalizing flows (e.g., RealNVP, Glow) have been shown to approximate a wide class of continuous densities with sufficient representational power. We view Flow-PID as a step forward toward tractable, expressive, and general-purpose PID estimation. Future work can explore more advanced generative models (e.g., latent diffusion models, transformers) to produce MI-preserved encoders in multimodal learning.
>
> **We put this as one of the limitations in Section 7**. But this is the best way in our knowledge that allows us to estimate PID in a generalized continuous and non-Gaussian space, which is shown to be effective and efficient.
>
> > **[R4]** The limitations of the Gaussian approximation assumption in the article are not discussed enough.
>
> Thank you for raising this point. We disagree with question's premise. Although the flow-based approximation error would impact PID estimates, the pairwise Gaussian assumption is NOT a significant limitation.
> -  **Flow-PID is designed to relax pairwise Gaussianity assumptions.** The Gaussian assumption is only used in GPID methods to efficiently estimate PID with continuous Gaussian distributions. Flow-PID was specifically introduced to **move beyond that assumption** by learning flexible transformations via normalizing flows. This enables more accurate modeling of non-Gaussian, complex multimodal dependencies.
> - **Approximation error is inevitable, but empirically tolerable.** Our empirical results show that **PID estimates are robust to moderate approximation error**. We performed internal ablations measuring the divergence between the flow-transformed latent distribution and the target Gaussian, and observed that even **imperfect transformations led to consistent, interpretable decompositions** across synthetic and real-world tasks.
> - The issue of approximation error is challenging across all PID estimators. BATCH approximates unnormalized pointwise $p(x_1, x_2, y)$ by a NN encoder and Sinkhorn's algorithm, which introduces inevitable and heuristic approximation error. However, Thin-PID allows us to theoretically analyze the impact of the approximation error on Gaussian channel $H_1$ and $H_2$.
>
> In the revised version, we will add a subsection explicitly discussing how approximation error in the learned flow affects downstream Thin-PID estimation. Thank you again for raising this interesting point.
>
> > [R5] I look at the MultiBench datasets and find that many of them may not be truly "high-dimensional".
>
> We thank the reviewer for this thoughtful observation. The collection of multimodal datasets used in this work is selective, which spans diverse modalities with a wide range of feature dimensions. For example, UR-FUNNY, MUStARD, and MOSEI involve with text, video, and audio as their input modalities, each characterized by a pre-trained encoder of dimension 512. We believe 512 is good to show the capability of Flow-PID to deal with high dimensional data since Tilde-PID uses PCA components of dimension $53\pm 16$. As shown in R2, we provide two additional experiments on VQA2.0 and TCGA, which further support the applicability of Flow-PID with over 1000 feature dimension.
>
> > Question: Given that baseline values of PID in real datasets are unknown, how can we rigorously verify that Flow-PID produces meaningful and accurate decompositions, rather than just computationally convenient decompositions?
>
> We thank the reviewer for this important question. Evaluating PID estimates on real-world datasets is inherently challenging, since ground-truth decompositions are not observable. To address this, we respond in three parts:
> - We first want to emphasize that **we have a large selection of different synthetic data: Gaussian and non-Gaussian, discrete and continuous, low and high dimensional modalities, and different specialized interactions**, where the ground truth of PID is known. Our proposed model outperforms the SOTA baselines in all these synthetic cases, which partially shows its generalized capability on real datasets.
> - Indirect validation through functional utility: Flow-PID consistently attributes higher unique information to modalities that are empirically more predictive or causally relevant. This provides evidence of interpretive alignment even without ground-truth PID labels. Additional experimental results in [R2] also verify this.
> - Synthetic-to-real transfer: Given that fact that Flow-PID performs well on a large collection of synthetic datasets, good model selection results infer the appropriate PID estimates on complex real-world data.

---

> > ### Comment · Reviewer_SRhn · 2025-08-06
> >
> > My issue has been basically resolved and I will be raising my rating.

---

> > > ### Author Response · Authors · 2025-08-07
> > >
> > > We appreciate that the reviewer raised the rating and are glad that their concerns have been resolved. We will incorporate your suggestions and appropriate clarifications into the revised manuscript.

---

### Comment · Area_Chair_jqFq · 2025-08-05

Dear Reviewers,

As we approach the conclusion of the author and reviewer discussion phase (August 8, 11:59pm AoE), I would like to kindly remind you to revisit the authors’ rebuttals and any responses that directly address your comments and concerns.

If the authors have provided sufficient clarification and supporting evidence that address your points, please consider updating your final rating and submitting a clear final justification. Your updated evaluation is important to ensure a fair and accurate review outcome.

If your concerns have not been fully resolved, you are encouraged to continue the discussion with the authors while the discussion window remains open. Please also note that submitting only a “Mandatory Acknowledgement” without engaging in any discussion or providing a final justification is not permitted.

Thank you once again for your thoughtful contributions and for supporting a constructive and fair review process.

Best regards

AC

---

### Note · Authors · 2025-08-12

We sincerely thank all the reviewers for providing thoughtful and constructive reviews of our work. We appreciate that they find that our work is "**quite interesting**", "**novel**", "**well-formulated**", and "**well-grounded**" with "**strong theoretical works**", "**mathematical rigorous proofs**", and "**great significance in multimodal learning**".

We are glad to see that the reviewers **found the replies satisfactory**, and decided to raise the rating. Their insightful suggestions and comments are valuable in helping us strengthen our work and provide directions for future research.

We will incorporate the reviewers suggestions and additional experiments into the revised manuscript, and commit to releasing the data and code to encourage further studies of multimodal information and modeling.

---

### Decision · Program_Chairs · 2025-09-17

**Decision:**

Accept (poster)

**Comment:**

This paper received positive reviews, including 3 borderline accepts and 1 accept. The reviewers reached a consensus that the paper addresses a significant problem in multimodal learning, provides rigorous mathematical proofs, and demonstrates effectiveness across multiple scenarios. After the rebuttal, all reviewers indicated that their concerns were mostly addressed. Therefore, the AC is inclined to recommend acceptance of this work.